# A Reality Check on Robust Bandit Algorithms for Buffer-Aware Early Exits

## Abstract

Early-exit neural networks (EENNs) reduce inference cost by allowing inputs to terminate at intermediate layers when classification confidence exceeds a threshold. However, practical deployments must operate under stochastic arrivals, limited device resources, and finite buffers, where the backlog directly impacts performance. This paper provides a systems-oriented study of buffer-aware EENNs and introduces new learning algorithms for threshold selection. First, we report results from real testbed experiments on heterogeneous devices, showing that incorporating buffer state into early-exit decisions substantially improves throughput and accuracy under load. Second, we extend policy gradient methods by integrating the Tsallis-softmax parameterization, which yields tunable exploration, robustness to high-variance rewards, and connects recent advances in the $\delta$-exponential family for policy optimization to practical scheduling in EENNs. Third, we propose contextual bandit algorithms that exploit the natural monotonic relationship between backlog and urgency via parametrized thresholds, reducing sample complexity and enabling generalization across system loads. Together, these contributions highlight that early exits are not only a model-design mechanism but also a systems scheduling problem, bridging theory and practice for robust and efficient inference in resource-constrained environments.

## 1 Introduction

Deep Neural Networks (DNNs) have achieved remarkable advances in performance in computer vision tasks (Krizhevsky et al., 2012; He et al., 2016; Edozie et al., 2025). However, their substantial computational demands hinder deployment on mobile devices for inference. A common workaround leverages cloud infrastructures equipped with GPUs (Satyanarayanan, 2017). In a cloud-only setting, mobile devices capture data and transmit it to the cloud, where the entire DNN inference is executed. The limitations of a cloud-only approach have motivated edge-cloud collaborative DNN inference. In this setup, edge devices process the initial layers of the model and send intermediate results to the cloud for further computation. This hybrid strategy reduces latency and improves resource utilization, overcoming the drawbacks of performing inference solely at the edge or in the cloud. By distributing the model across heterogeneous resources, the system benefits from the strengths of both environments. Nonetheless, this approach demands offloading to the cloud, which introduces communication and energy overheads, without ensuring higher accuracy. Early-exit neural networks (EENNs) provide a promising solution to mitigate these overheads while maintaining flexible edge-cloud inference (Kang et al., 2017; Hu et al., 2019; Kag & Fedorov, 2023).

EENNs exploit the observation that the initial neural layers can produce confident predictions for many inputs. To enable this, EENNs have exit branches (i.e., intermediary classifiers) inserted at their intermediate neural layers. For each input, these exit branches can estimate their classification confidence. If it exceeds a predefined threshold, the input can be classified early at the exit branch. Otherwise, it continues through the remaining layers for further processing (Teerapittayanon et al., 2016; Bajpai & Hanawal, 2025). Therefore, EENNs have emerged as a powerful approach to reduce inference latency and energy consumption in deep models by allowing inputs to exit early once classification confidence surpasses a threshold (Liu et al., 2023; Mishra et al., 2025). While most prior work has focused on algorithmic aspects of early exits (e.g., learning confidence thresholds (Pacheco et al., 2024) or optimizing placement of side branches (Ju et al., 2021)), the integration of early exits into *real systems* remains underexplored. In practice, inputs arrive according to stochastic

processes, devices operate under heterogeneous hardware constraints, and buffering is unavoidable when workloads exceed instantaneous processing capacity (Chen et al., 2025). These system-level realities motivate a rethinking of how early exits should be managed in realistic deployments.

In this paper, we make three main contributions:

**A systems perspective on buffer-aware early exits.** We present results from real experiments conducted on a heterogeneous testbed consisting of a Raspberry Pi and a MiniPC, where images are generated at the edge and either classified locally via an early exit or forwarded to the powerful device, acting as a cloud server, for deeper processing. Crucially, we account for the impact of the *buffer state*, i.e., the number of pending jobs in the queue, on the early-exit decision. Our experiments illustrate how buffer-aware scheduling improves both throughput and accuracy under varying load, offering the first empirical evidence of how queue dynamics shape early-exit efficiency in practice (Sections 2 and 4).

**Policy gradient with Tsallis-softmax thresholds.** We extend classical policy gradient algorithms for threshold selection by incorporating the *Tsallis-softmax parameterization* (Zhu et al., 2025). This design choice builds robustness into the system, as Tsallis-softmax policies yield tunable exploration, heavier-tailed action distributions, and improved stability under high-variance rewards. We connect recent advances on the *$\delta$-exponential family for policy optimization* with the concrete problem of buffer-aware early exits, demonstrating how insights from reinforcement learning theory can be translated into practical scheduling gains (Section 3.1).

**Parametrized Upper Confidence Bound (UCB) with monotonic thresholds.** We introduce a parametrized, monotonic threshold function of the form $\alpha(q) = \theta_1 q + \theta_2$, where $q$ is the backlog size. This approach leverages domain knowledge by explicitly capturing the natural *monotonic relationship between backlog and urgency*: as queues grow longer, early exits should become more likely. Embedding this structure into the learning process reduces the memory footprint, improves robustness to noise, and accelerates convergence. Importantly, it enhances the algorithm's ability to generalize across varying system loads with fewer samples, making it suitable for deployment resource-constrained environments (Section 3.2).

These contributions establish a new perspective on early-exit neural networks: one that integrates system dynamics, leverages robust reinforcement learning primitives, and introduces structured bandit algorithms. By grounding our work in both *real testbed experiments* and *theoretical extensions*, we demonstrate that early exits are not only a model-design choice, but also a *systems scheduling problem* — and that addressing this perspective is key to unlocking their full potential in practice.

**Outline.** Section 2 introduces EENNs and the queue-aware objective and Section 3 presents the two considered learning approaches (policy gradient and UCB). Section 4 details our heterogeneous testbed and reports results. Section 5 reviews related work and Section 6 concludes. Appendices A to G contain formal results, supplementary material and a link to our anonymized code repository.

## 2 EARLY-EXIT NEURAL NETWORKS

Early-exit Deep Neural Networks (EENNs) augment a backbone model (e.g., MobileNetV2 (Dong et al., 2020)) by adding one or more *exit branches*, which enable inputs to exit the model early once sufficient confidence is reached. This mechanism enables an edge-cloud collaborative DNN inference. Moreover, it allows a flexible trade-off between inference latency and predictive accuracy, making EENNs particularly suited for resource-constrained or time-sensitive applications.

An EENN processes an input $\mathbf{x}$ layer by layer, with intermediate branches that can output a prediction before the final layer. At an intermediate exit, the model computes a probability vector $\mathbf{p}_I(\mathbf{x})$ and its confidence $C_I(\mathbf{x}) = \max_c [\mathbf{p}_I(\mathbf{x})]_c$. If $C_I(\mathbf{x}) \geq \alpha$, where $\alpha \in [0, 1]$ is a threshold, the sample is classified locally with prediction $\hat{y}_I(\mathbf{x})$. Otherwise, it is forwarded (with overhead $o$) to the final layer, which produces probability vector $\mathbf{p}_L(\mathbf{x})$, confidence $C_L(\mathbf{x})$, and prediction $\hat{y}_L(\mathbf{x})$. For clarity, we omit the explicit dependence on $\mathbf{x}$ when unambiguous. Further details on notation and background for EENNs are provided in Appendices A and B, respectively.

**Why a learned threshold?** The optimal $\alpha$ depends on *context*. Even for a fixed model and data distribution, the preferred trade-off between accuracy and latency shifts with load (e.g., backlog, device heterogeneity, or network contention). A fixed threshold is inherently fragile. We instead *learn* thresholds online using bandit algorithms, viewing each candidate threshold as an action.

## 2.1 A BANDIT VIEW OF EARLY EXITS

In a multi-armed bandit (MAB) setup, each action is associated with an unknown instantaneous reward distribution. In this paper, the reward $r_t$ is a function of the selected action (threshold) $\alpha_t$. When an early exit is taken, no further improvement in confidence can be obtained, resulting in a reward of zero. Otherwise, if the input proceeds through the final layers, the reward corresponds to the confidence gain from the intermediate to the final layer, denoted as $\Delta C = \max(C_L - C_I, 0)$, discounted by the additional overhead of data offloading $o_t$. Hence, the reward $r_t(\alpha_t)$ can be written as

$$r_t(\alpha_t) = \begin{cases} 0, & \text{if } C_I \geq \alpha_t \quad \text{(early exit)}, \\ \Delta C - o_t, & \text{otherwise}, \end{cases} \tag{1}$$

where $\Delta C$ is an *accuracy-gain proxy*: it is large when the final layer is confidently more accurate than the side branch, and zero when the side branch was already as confident as the tail of the EENN model (Bajpai & Hanawal, 2025; Casale & Roveri, 2023; Pacheco et al., 2024). The term $o_t$ is the overhead incurred by processing up to the last layer. We assume that it ranges between $[0, 1]$, making $\Delta C$ and $o_t$ commensurable. The overhead may stem from communication latency when offloading data to the cloud, from energy consumption due to additional processing, or from both factors combined. In this work, we instantiate the overhead as a linear function of the backlog $q_t$ ensuring that the penalty remains on the same scale as the confidence gains, i.e., $o_t(q_t)$ is in the unitary interval for the feasible backlog sizes.

Equivalently (up to the action-independent baseline $C_I$) the reward could take the more symmetric form

$$\tilde{r}_t(\alpha_t) = \begin{cases} C_I, & \text{if the sample exits early}, \\ C_L - o(q_t), & \text{otherwise}, \end{cases}$$

so both rewards $r_t$ and $\tilde{r}_t$ induce the same optimal policy and regret. We emphasize that the backlog is a genuine state variable that modulates the relative advantage of early exit versus full processing through $o(q_t)$: when $q_t$ is small, $o(q_t)$ is small and full processing is favoured; as $q_t$ grows, $o(q_t)$ increases and early exits become preferable.

**Expected reward.** At each round $t$, a context $q_t \in \mathcal{Q} = \{0, \ldots, Q_{\max}\}$ is observed, an action $\alpha_t \in \mathcal{A}$ is chosen, and a reward $r_t \in [0, 1]$ is observed. For each pair $(q, \alpha) \in \mathcal{Q} \times \mathcal{A}$, there is an unknown distribution $\nu_{q,\alpha}$ with mean $\mu(q, \alpha) := \mathbb{E}_{r \sim \nu_{q,\alpha}}[r] = \mathbb{E}[\Delta C - o \mid C_I < \alpha] \cdot P[C_I < \alpha]$, since $r(\alpha) = (\Delta C - o(q)) \mathbf{1}_{\{C_I < \alpha\}}$, where $P[C_I < \alpha]$ represents the probability of processing up to the last layer. The controller's objective is to find an optimal threshold $\alpha^*(q) \in \arg\max_{\alpha \in \mathcal{A}} \mu(q, \alpha)$ that maximizes the expected reward $\mu(q, \alpha)$.

**Policy and regret.** A policy specifies how the controller chooses a threshold $\alpha_t$ at round $t$, given the reward history observed up to that point. To evaluate a given policy, we define the expected regret $R_T := \sum_{t=1}^{T} \left( \mu(q_t, \alpha^\star(q_t)) - \mu(q_t, \alpha_t) \right)$, up to the horizon of $T$ rounds, which quantifies the cumulative expected cost of making suboptimal decisions. The expected regret grows over time as the controller balances exploration (testing different thresholds) and exploitation (choosing the currently best-performing threshold). A difficulty here is the fact that the backlog length $q_t$ is *endogenous*: it evolves according to the queueing dynamics and is directly affected by the chosen exits (actions), so that both the context process and the reward distributions become action-dependent and potentially non-stationary. A full regret analysis in this stateful, queue-dependent setting would require a more elaborate treatment (closer to Markovian bandits or simple MDPs with queueing dynamics), which is beyond the scope of this paper. We present Theorem C.1 (Appendix) as an illustrative regret result that clarifies how our UCB-based policy fits into the classical stochastic contextual bandit framework under a discretized, exogenous-queue approximation.

**Buffer-aware overhead.** In realistic deployments, inputs may accumulate in a queue before being processed by the EENN model at the edge device. Let $q_t = |Q_t|$ denote the *backlog* at decision time; a larger $q_t$ implies a longer queuing delay and a higher risk of drops when the buffer capacity $B$ is tight. We model the overhead as a queue-aware penalty,

$$o_t \equiv o(q_t) = \nu q_t - \kappa, \tag{2}$$

with $\nu \geq 0$ and $\kappa \geq 0$. Plugging equation 2 into equation 1 yields a reward that balances *accuracy gain* against *latency pressure*. Therefore, rewards are queue-aware: if the intermediate confidence $C_I$

exceeds the sampled threshold $\alpha_t$, the sample exits early with reward $r_t = 0$. Otherwise, it proceeds to the final layer, and the reward is

$$r_t = \max(C_L - C_I, 0) - (\nu q - \kappa), \tag{3}$$

where the first term $\max(C_L - C_I, 0)$ measures the *information gain* from continuing deeper in the network, while the second term $(\nu q - \kappa)$ penalizes queue buildup, balancing accuracy improvements against system congestion. In this way, the policy learns to adapt thresholds $\alpha_t$ not only to prediction confidence but also to the current backlog, ensuring accuracy and timeliness in dynamic workloads. The affine choice $o(q_t) = \nu q_t - \kappa$ is just a modelling simplification used in our experiments; the algorithms only require a bounded reward, and can in principle handle more general monotone penalties (steeper affine, convex, or SLA-style piecewise functions).

## 2.2 FROM VANILLA TO ROBUST, BUFFER-AWARE LEARNING

The bandit perspective supports two complementary families we study in Section 3.

**Vanilla bandits.** We begin with two standard algorithms that treat each threshold $\alpha \in \mathcal{A}$ as an independent arm. In the *vanilla policy-gradient* variant (Algorithm 1 with $\delta = 1$), each backlog state $q$ maintains an independent softmax distribution over $\mathcal{A}$, parameterized by $\{\theta_q(\alpha)\}$. In the *vanilla UCB* variant, the controller keeps per-threshold estimates $\widehat{Q}(\alpha)$ and counts $N(\alpha)$, and selects

$$\alpha_t \in \arg\max_{\alpha \in \mathcal{A}} \widehat{Q}(\alpha) + \beta\sqrt{\frac{\ln t}{N(\alpha) + 1}}, \tag{4}$$

with on-policy updates from $r_t(\alpha_t)$ (Algorithm 2). Under the simplest setting, previous information about the structure of the optimal policy, e.g., thresholds should decrease with backlog, is not taken into account.

**Robust bandits with structure.** We next consider two enhancements that improve robustness and sample efficiency. First, *Tsallis exploration*: replacing the softmax with the $\delta$-softmax from the generalized $\delta$-exponential family (Algorithm 1, $\delta \neq 1$) yields heavier- or lighter-tailed action distributions with a single knob $\delta$, stabilizing learning under high-variance, queue-induced rewards. Second, *monotone, parametric thresholds*: instead of learning a separate threshold for each backlog state, we encode domain knowledge that *urgency increases with backlog* by restricting policies to monotone mappings

$$\alpha(q) = \mathrm{clip}_{[0,1]}\big(\theta_1 q + \theta_2\big), \qquad \theta_1 \leq 0, \tag{5}$$

and apply contextual UCB over $\theta = (\theta_1, \theta_2)$ (Algorithm 2) or policy-gradient over the induced discrete $\mathcal{A}$. This structural bias reduces memory, accelerates convergence, and promotes generalization across backlog states.

**Relation to Bandits with Knapsacks and constrained MDPs.** As requested by a reviewer, we clarify our positioning with respect to Bandits with Knapsacks (BwK) (Badanidiyuru et al., 2018) and budgeted bandits (Madani et al., 2004; Cayci et al., 2019; Xia et al., 2015). Conceptually, our controller also trades off accuracy against resource usage (computation and backlog), so there is a clear connection. Formally, however, classical BwK/budgeted bandits operate with a monotone, non-renewable budget that is consumed by arm pulls and the process stops when the budget is exhausted. In our model, the backlog $q_t$ is a queueing state that evolves with arrivals and departures, can both increase and decrease, and is bounded by a finite buffer; the "resource" is instantaneous buffer capacity in a steady-state regime, not a depleting budget in a finite-horizon regime. Moreover, we do not perform primal–dual optimization over Lagrange multipliers: the penalty $o(q_t)$ is a modelling choice (reward shaping) rather than a dual variable. The existing BwK algorithms do not apply directly to our setting without substantial modification.

**Putting it together.** The combination of queue-aware rewards (see equation 1 and equation 2) with (i) robust exploration via $\delta$-softmax and (ii) monotone threshold parameterizations model the trade-off between accuracy and latency. Under light load (small $q$), the learned policy tends to defer to the final layer when the expected gain $\mathbb{E}[\max(C_L - C_I, 0)]$ exceeds the small penalty $o(q)$. Under heavy load, the penalty dominates and the policy shifts toward earlier exits, stabilizing throughput and reducing loss. Our experiments in Sec. 4 validate these behaviors on a heterogeneous Raspberry Pi $\rightarrow$ MiniPC testbed across stationary and non-stationary workloads.

---

**Algorithm 1:** Policy Gradient for Queue-Aware EENNs with $\delta$-Softmax Thresholds

---

**1 Input:** learning rate $\eta$, max queue $Q_{\max}$, Tsallis parameter $\delta$
**2 Init:** $\theta_q(\alpha) \leftarrow 0$ for all $q \in \{0, \ldots, Q_{\max}\}$ and $\alpha \in \mathcal{A}$
**3 for** $t = 1, 2, \ldots, T$ **do**
**4**      Observe next event (arrival or service); update backlog $q \leftarrow |Q|$ ;      # Track queue size
**5**      Compute policy over $\mathcal{A}$:
$$\pi_t^{(q)}(\alpha) \;=\; \exp_\delta\big(\theta_q(\alpha)\big) \Big/ \sum_{\alpha' \in \mathcal{A}} \exp_\delta\big(\theta_q(\alpha')\big)$$
     Sample threshold $\alpha_t \sim \pi_t^{(q)}$ ;      # Select action (confidence threshold)
**6**      Obtain $C_I$;
**7**      **if** $C_I \geq \alpha_t$ **then**
**8**          $r_t \leftarrow 0$ ;      # Early exit
**9**      **else**
**10**          Obtain $C_L$
**11**          $r_t \leftarrow \max(C_L - C_I, 0) - (\mu q - \kappa)$ ;      # Accuracy gain minus queue penalty
**12**      Update baseline $B_t$;
**13**      **foreach** $\alpha \in \mathcal{A}$ **do**
**14**          $g_t(\alpha) \leftarrow \big(\mathbb{1}\{\alpha = \alpha_t\} - \pi_t^{(q)}(\alpha)\big) \big/ \big(1 + (1 - \delta)\theta_q(\alpha)\big)$ ;      # Tsallis component
**15**          $\theta_q(\alpha) \leftarrow \theta_q(\alpha) + \eta\,(r_t - B_t)\,g_t(\alpha)$ ;      # Gradient ascent update

---

## 3 ROBUST BANDITS

We now turn to bandit-based methods for learning queue-aware thresholds in early-exit neural networks. We present two complementary approaches: *policy-gradient* methods (Section 3.1) and *UCB-based* methods (Section 3.2). For each of these approaches, we distinguish between a *vanilla* case and a *robust* case. In the vanilla setting, policies are learned using standard formulations (softmax for policy gradient, tabular thresholds for UCB). In the robust setting, we incorporate additional structure or parameters: for policy gradient, the $\delta$-softmax generalization (Tsallis-softmax) provides tunable exploration and robustness to high-variance rewards; for UCB, a monotonic parameterization $\alpha(q) = \theta_1 q + \theta_2$ captures the natural relationship between backlog and urgency, reducing sample complexity and improving generalization. This extends the flexibility of vanilla methods to introduce robustness and system-awareness into bandit-based threshold selection, whose experimental evaluation in a testbed is reported in Section 4.

### 3.1 POLICY-GRADIENT WITH $\delta$-SOFTMAX THRESHOLDS

We study a policy-gradient approach for learning queue-aware thresholds in EENNs. At each step, the system observes the backlog $q = |Q|$ and samples a threshold $\alpha_t \in \mathcal{A}$ according to a $\delta$-softmax policy. This policy is based on a deformation of the exponential function, introduced in Tsallis (1988), defined as

$$\exp_\delta(x) = \begin{cases} \exp(x), & \delta = 1, \\ \big(1 + (1-\delta)x\big)^{\frac{1}{1-\delta}}, & \delta \neq 1, \ (1-\delta)x > -1. \end{cases}$$

The special case $\delta = 1$ recovers the standard softmax, while $\delta \neq 1$ yields Tsallis-softmax with tunable exploration. From a statistical viewpoint, the inverse function $\log_\delta x := \exp_\delta^{-1}(x)$ (Appendix D), known as the Tsallis $\delta$-logarithm, coincides with the classical Box–Cox power transform with parameter $\lambda = 1 - \delta$, widely used for variance stabilization and heavy-tail mitigation in statistics.

**Vanilla policy-gradient.** In the vanilla case, the policy at each backlog state $q$ is parameterized by $\theta_q(\alpha)$ for all $\alpha \in \mathcal{A}$, and action probabilities are computed using the standard softmax ($\delta = 1$). This setup is straightforward and widely used, but it lacks flexibility: the exploration–exploitation trade-off is entirely determined by the scale of the parameters, and the resulting distributions often assign non-negligible probability mass even to poor actions.

**Robust policy-gradient.** In the robust case, we generalize to the $\delta$-softmax family, which yields Tsallis-softmax policies. The additional parameter $\delta$ provides explicit control over exploration: for $\delta < 1$, the policy explores more broadly, while for $\delta > 1$ the distribution becomes sharper. This tunability introduces several advantages: it improves robustness under high-variance rewards, yields heavier-tailed action distributions that avoid premature convergence, and allows tailoring the degree of exploration to system load. By bridging standard softmax with Tsallis-softmax, the robust policy-gradient extends the design space for queue-aware early-exit scheduling. Appendix D gives the general derivation of the gradient-ascent update used at line 14 of Algorithm 1. Despite the growing interest in the $\delta$-exponential family (e.g., (Zhu et al., 2025)), to the best of our knowledge this specific gradient-ascent rule for $\delta$-softmax (Tsallis-softmax) policies is novel to this work.

**Generalized softmax family ($\delta$-softmax).** Our robust policy-gradient design builds on a generalized exponential family that has appeared in several guises across the literature—including the Box–Cox transform (Box & Cox, 1964), the $\delta$-exponential/Tsallis family (Tsallis, 1988; Zhang & Sabuncu, 2018), and $\alpha$-fairness (Kelly, 1997). Though introduced in different domains, these formulations share a common shape control that smoothly trades off exploration sharpness and tail heaviness. Recent work has revisited this family for reinforcement learning, showing stability and performance gains in policy optimization (Zhu et al., 2025). In this paper, we take advantage of a Tsallis-softmax ($\delta$-softmax) policy for threshold selection in buffer-aware early exits, yielding tunable exploration, robustness to queue-induced variance, and an algorithm for buffer-aware early exits, unifying stability and adaptability in dynamic edge workloads.

## 3.2 UCB

Algorithm 2 is written at a high level over a *policy class* $\mathcal{H}$, where each policy $h \in \mathcal{H}$ maps the current backlog $q$ to a threshold $\alpha = h(q)$. This abstraction allows us to unify different design choices. In our work, we focus on two concrete instantiations:

**Vanilla UCB.** Here, $\mathcal{H}$ contains one policy for each backlog value $q \in \{0, \ldots, Q_{\max}\}$ and threshold $\alpha \in \mathcal{A}$. That is, $h(q)$ is constant in $q$ and simply represents the choice of a fixed threshold for a particular backlog state. This corresponds to learning an independent threshold for each backlog size, without imposing any structure across queue states. While flexible, this approach requires storing and exploring $|\mathcal{A}| \cdot Q_{\max}$ arms and thus scales poorly as $Q_{\max}$ grows.

**Robust UCB.** To exploit domain knowledge, we can instead define a structured class

$$\mathcal{H} = \{h_\theta(q) = \mathrm{clip}_{[0,1]}(\theta_1 q + \theta_2) \,:\, \theta = (\theta_1, \theta_2) \in \Theta, \ \theta_1 \leq 0\}. \tag{6}$$

Here, the threshold is a linear function of backlog size $q$, with slope $\theta_1$ and intercept $\theta_2$. The constraint $\theta_1 \leq 0$ enforces a natural monotonicity: as the backlog grows, the system becomes more aggressive in taking early exits. This parametrization drastically reduces the number of candidate policies, encourages generalization across queue states, and accelerates convergence. Note that (6) is a particular choice – Algorithm 2 permits other forms of policy spaces.

Together, these two instantiations illustrate a spectrum between *flexibility* (tabular thresholds) and *structure* (monotonic parametrization). Our experiments show that incorporating monotonicity yields significant benefits in robustness and sample efficiency, particularly under high load.

**Why Tsallis-PG in addition to UCB?** Our goal was not to replace UCB but to complement it in a regime where its exploration can become inefficient. In our setting, each threshold $\alpha$ induces a distinct action, and a naive bandit discretisation leads to a large (potentially very large) number of arms. While UCB enjoys strong regret guarantees in this finite-armed setting, its exploration behaviour scales with the number of actions: when there are many candidate thresholds, UCB tends to spend a long time exploring suboptimal arms before concentrating on the best one, which can hurt its finite-sample efficiency in our high-resolution threshold regime.

In contrast, our Tsallis policy-gradient method operates over a parametric policy on a continuous (or finely discretised) threshold space and updates the policy in a more greedy, directionally informed way. Rather than treating each threshold as an independent arm, the gradient step shifts probability mass towards regions of thresholds that currently appear promising, which can significantly reduce the amount of uniform exploration required when the action space is large.

---

**Algorithm 2:** Contextual-UCB for Queue-Aware EENNs with Threshold Policies

---

1 **Input:** policy class $\mathcal{H}$, UCB parameter $\beta > 0$, max queue $Q_{\max}$
2 **Init:** $N(h) \leftarrow 0, \hat{Q}(h) \leftarrow 0$ for all $h \in \mathcal{H}$
3 **for** $t = 1, 2, \ldots, T$ **do**
4      Observe next event and set context $q \leftarrow |Q|$ ;              # Backlog as context
      # Select policy by UCB on its induced threshold at context $q$
5

$$h_t \leftarrow \arg\max_{h \in \mathcal{H}} \left\{ \hat{Q}(h) + \beta\sqrt{(\ln t)/(N(h)+1)} \right\}$$

     $\alpha_t \leftarrow h_t(q)$ ;                    # Threshold for current backlog
6      Obtain $C_I$ (and $C_L$, if no early exit) and compute reward $r_t$;
      # Update policy statistics
7      $N(h_t) \leftarrow N(h_t) + 1$;
8      $\hat{Q}(h_t) \leftarrow \hat{Q}(h_t) + \frac{r_t - \hat{Q}(h_t)}{N(h_t)}$;

---

## 4 EVALUATION

We evaluate buffer-aware EENNs under three settings: a stationary workload, a mixed *easy-hard* workload, and a mixed *hard-easy* workload. In *stationary workload*, images from the dataset are unaltered, with no image distortion, such as blur and noise. In the *easy–hard workload*, the arrival rate remains fixed, but while the first half of the experiment uses original images, the second half introduces images with gaussian blur. Finally, the *hard–easy workload* reverses the order of the *easy–hard workload*. For each case, we report both system-level performance and per-machine breakdowns. The metrics include accuracy, loss rate, loss ratio, score and goodput,

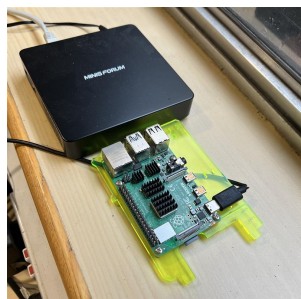

Figure 1: The edge-cloud testbed

$$Score = Accuracy - LossRatio, \quad Goodput = Accuracy \cdot ArrivalRate \cdot (1 - LossRatio),$$

where score and goodput combine accuracy and loss ratio to provide unified measures capturing the trade-off between accuracy and loss. Table 1 presents the results for these metrics. For completeness, Appendix E provides additional metrics, including arrival rate, throughput, and utilization.

**Testbed.** Our experimental setup combines a heterogeneous edge–cloud testbed with EENNs. The testbed, as shown in Figure 1, consists of two devices: a Raspberry Pi 4B Rev 1.4 with a buffer of size 10, and a MinisForum Z83-F equipped with an Intel Atom x5-Z8350 CPU and a buffer size of 50. The MinisForum is considered a MiniPC and serves as the cloud. The Raspberry Pi and MiniPC are connected via WLAN with a bandwidth of 30 Mbps. Jobs first arrive at the Raspberry Pi; if an input does not satisfy the early-exit condition at the intermediate branch, it is forwarded to the MiniPC for deeper processing.

**Model and dataset.** The EENN used in the experiment is based on *ResNet50*, with an intermediate exit branch placed at the 9th residual block (out of a total of 20). This design strikes a balance between intermediate accuracy and computation savings. The EENN is trained on *CIFAR-10* image classification dataset (Krizhevsky et al., 2009) with 60,000 images across 10 classes. A subset of 5,000 images from the test set is used as the workload in our experiment.

**Routing setup.** Images are generated on the Raspberry Pi. If the intermediate confidence $C_I$ exceeds the threshold $\alpha$, classification occurs locally and inference terminates. Otherwise, the sample is transmitted via WLAN to the MiniPC, which processes it until the final layer and outputs the classification result. The model deployed on the MiniPC does not contain any early-exits mechanisms.

**Scheduler settings.** The threshold selection scheduler is deployed on the Raspberry Pi. For the UCB, vanilla policy gradient (PG), and Tsallis policy gradient (TSA-PG) schedulers, the threshold arm $\alpha$ is selected from $\{0.50, 0.51, \ldots, 0.99\}$. For the monotonic-parameterized UCB (UCB-Mono), the action is $\boldsymbol{\theta} = (\theta_1, \theta_2)$ with $\theta_1 \in \{0.00, 0.01, \ldots, 0.05\}$ and $\theta_2 \in \{0.5, 0.51, \ldots, 0.99\}$. Samples that do not exit early on the Raspberry Pi are forwarded to the MiniPC, which always

Table 1: System-level results across workloads.

| Workload / Method | Accuracy | Loss rate | Loss ratio | Score | Goodput |
|---|---|---|---|---|---|
| **Stationary** | | | | | |
| Baseline | $0.8852 \pm 0.0011$ | $0.7865 \pm 0.0165$ | $0.1981 \pm 0.0031$ | $0.6871 \pm 0.0036$ | $2.8394 \pm 0.0115$ |
| UCB | $0.8705 \pm 0.0012$ | $0.3583 \pm 0.0139$ | $0.0902 \pm 0.0030$ | $0.7803 \pm 0.0031$ | $3.1679 \pm 0.0113$ |
| UCB-Mono | $0.8817 \pm 0.0011$ | $0.2902 \pm 0.0132$ | $0.0730 \pm 0.0030$ | $0.8087 \pm 0.0029$ | $3.2693 \pm 0.0113$ |
| PG | $0.8743 \pm 0.0013$ | $0.2053 \pm 0.0085$ | $0.0517 \pm 0.0020$ | $0.8227 \pm 0.0018$ | $3.3164 \pm 0.0086$ |
| TSA-PG(1.75) | $\mathbf{0.8682 \pm 0.0014}$ | $\mathbf{0.1488 \pm 0.0093}$ | $\mathbf{0.0374 \pm 0.0023}$ | $\mathbf{0.8308 \pm 0.0021}$ | $\mathbf{3.3429 \pm 0.0096}$ |
| **Easy–Hard** | | | | | |
| Baseline | $0.7829 \pm 0.0015$ | $0.8955 \pm 0.0169$ | $0.2256 \pm 0.0030$ | $0.5573 \pm 0.0035$ | $2.4251 \pm 0.0105$ |
| UCB | $0.7739 \pm 0.0019$ | $0.5368 \pm 0.0146$ | $0.1351 \pm 0.0030$ | $0.6387 \pm 0.0036$ | $2.6774 \pm 0.0114$ |
| UCB-Mono | $0.7818 \pm 0.0019$ | $0.4391 \pm 0.0147$ | $0.1105 \pm 0.0032$ | $0.6713 \pm 0.0035$ | $2.7816 \pm 0.0121$ |
| PG | $0.7733 \pm 0.0019$ | $0.3257 \pm 0.0114$ | $0.0819 \pm 0.0026$ | $0.6914 \pm 0.0025$ | $2.8399 \pm 0.0106$ |
| TSA-PG(1.75) | $\mathbf{0.7682 \pm 0.0020}$ | $\mathbf{0.2619 \pm 0.0143}$ | $\mathbf{0.0659 \pm 0.0034}$ | $\mathbf{0.7023 \pm 0.0033}$ | $\mathbf{2.8703 \pm 0.0128}$ |
| **Hard–Easy** | | | | | |
| Baseline | $0.7829 \pm 0.0023$ | $0.9045 \pm 0.0173$ | $0.2278 \pm 0.0031$ | $0.5551 \pm 0.0036$ | $2.4182 \pm 0.0120$ |
| UCB | $0.7728 \pm 0.0022$ | $0.5074 \pm 0.0171$ | $0.1277 \pm 0.0037$ | $0.6450 \pm 0.0037$ | $2.6965 \pm 0.0138$ |
| UCB-Mono | $0.7825 \pm 0.0023$ | $0.4408 \pm 0.0153$ | $0.1109 \pm 0.0033$ | $0.6716 \pm 0.0028$ | $2.7829 \pm 0.0132$ |
| PG | $0.7736 \pm 0.0021$ | $0.3223 \pm 0.0095$ | $0.0811 \pm 0.0021$ | $0.6925 \pm 0.0024$ | $2.8434 \pm 0.0101$ |
| TSA-PG(1.75) | $\mathbf{0.7687 \pm 0.0027}$ | $\mathbf{0.2672 \pm 0.0164}$ | $\mathbf{0.0672 \pm 0.0039}$ | $\mathbf{0.7015 \pm 0.0032}$ | $\mathbf{2.8682 \pm 0.0157}$ |

processes them to completion. We instantiate the queue-aware overhead using the affine form $o(q) = 0.1q - 0.05$. For the considered backlog sizes ($1 \leq q \leq 10$), this choice ensures that the overhead $o(q)$ is commensurable with the confidence gains and lies in the interval $[0, 1]$.

**Arrival process and duration.** The input images arrive at the Raspberry Pi according to a Poisson process with rate 4 requests per second. This ensures that the system operates in a moderately loaded regime: high enough to induce queueing effects, yet low enough to allow meaningful differentiation between methods without saturating the MiniPC worker. Each experiment runs for one hour, providing sufficient samples to evaluate steady-state performance in terms of accuracy, throughput, and queueing metrics.

Although our empirical setting is limited (single mid-network exit, CIFAR-10, Poisson arrivals) we clarify that the *modeling layer*—the queue-aware reward and the buffer-aware bandit formulation—is not tied to these choices. The same queue-dependent objective and learning rules can be applied to multi-exit architectures (by parameterizing thresholds per exit) and to other datasets and backbones, and Poisson arrivals can be replaced by more realistic arrival processes at the cost of more complex queue dynamics. Our goal here is to establish and analyze the queue-aware bandit formulation and to show that it can be implemented end-to-end on a real testbed.

### 4.1 GENERALIZED-EXPONENTIAL POLICY GRADIENT OFFERS FLEXIBLE SCHEDULING

First, we evaluate the methods under the stationary workload. Table 1 presents in its upper part the system-level metrics under a stationary arrival process. Buffer-aware variants (UCB) significantly reduce the loss ratio by $54.5\%$ compared to the original baseline. The monotonic parameterization of UCB further improves the score from $0.7814$ to $0.7842$. Policy-gradient-based methods offer further improvements, resulting in significantly lower loss ratios. In particular, the policy gradient method with Tsallis-softmax ($\delta = 1.75$) achieves the lowest loss ratio ($0.0371 \pm 0.0013$), which is $83.4\%$ lower than the baseline, and the highest score ($0.8312 \pm 0.0015$), which is $26.1\%$ higher than the baseline, among all evaluated methods.

Next, we evaluate the methods under context drift workloads. In the easy-hard setting, the first 30 minutes consist of original CIFAR10 images, referred to as easy jobs. In the subsequent 30 minutes, a Gaussian blur with a kernel size of 5 and a sigma of 3 is applied to the images, and these images are hard jobs. Similarly, in the hard-easy setting, the workload begins with blurred images (hard jobs) for the first 30 minutes, followed by original images (easy jobs) for the next 30 minutes.

Table 1 reports the results for the stationary, easy-hard and hard-easy workloads. In both dynamic workloads, all methods experience a reduction in accuracy and an increase in loss ratio as jobs become more challenging to classify, requiring deeper layers of processing to achieve higher confidence scores. All proposed variants significantly outperform the baseline method by at least $14.5\%$ and $16.2\%$ in easy-hard and hard-easy settings, respectively. Among them, the Tsallis-softmax policy gradient with $\delta = 1.75$ consistently balances classification accuracy and loss ratio well, achieving the highest performance scores of $0.7023 \pm 0.0033$ and $0.7015 \pm 0.0032$ in the easy-hard and hard-easy workloads, respectively.

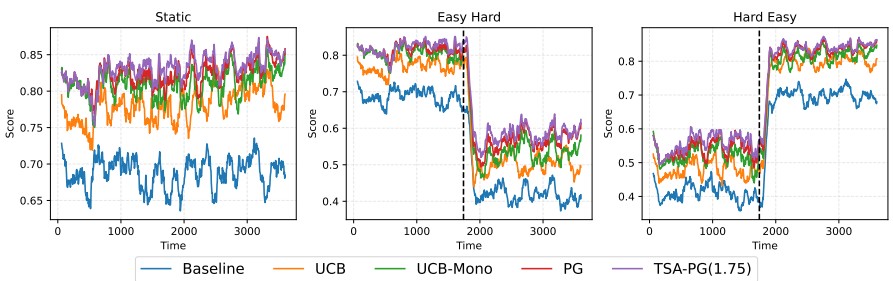

Figure 2: Monitored performance score over time

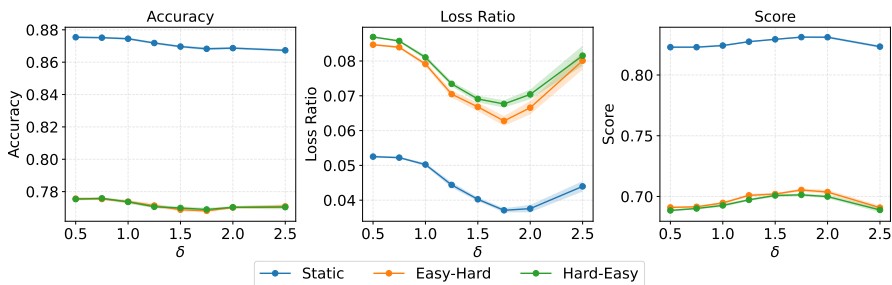

Figure 3: Sensitivity analysis on various $\delta$ used in the Tsallis-softmax policy gradient

Figure 2 presents the monitored system performance score over time across three types of workloads. Under the stationary workload, the performance score of PG and TSA-PG (1.75), which measures the trade-off between accuracy and system loss ratio, presents an increasing trend. This indicates that policy-gradient methods progressively learn to generate thresholds that improve system performance score. In the easy–hard and hard–easy workloads (with the black vertical line indicating the context change), the performance score experiences a sharp increase or decrease after the change is introduced. The policy-gradient methods adapt to the new context, and the performance score increases again over time. In contrast, this adaptive effect is less obvious for UCB-based methods.

## 4.2 Sensitivity analysis

A sensitivity analysis is conducted to examine the impact of $\delta \in [0.5, 2.5]$ in the Tsallis-softmax policy gradient. As shown in Figure 3, the loss ratio decreases significantly as $\delta$ increases, reaching its minimum at $\delta = 1.75$. Meanwhile, accuracy experiences only a slight decline. The performance score indicates that the Tsallis-softmax policy gradient achieves the best accuracy–loss trade-off at $\delta = 1.75$. However, when $\delta$ exceeds 1.75, the loss ratio increases and the performance score decreases, indicating that the algorithm is over-exploiting and has become trapped in solutions that have already been explored.

Across all settings, we observe three consistent trends: (i) buffer-awareness variant improves robustness compared to the original baseline; (ii) policy gradient methods with Tsallis-softmax outperform their vanilla counterparts by providing tunable exploration; and (iii) the $\delta = 1.75$ configuration offers the best trade-off between accuracy and queue stability.

**Takeaway.** In our setting, we discretize the parameter space into as many as 100 thresholds, which goes beyond the small discrete spaces where UCB is typically most effective. While UCB-based algorithms remain appealing due to their theoretical regret guarantees and robustness in noisy environments, their efficiency decreases as the discretization grows. Policy gradient methods, by contrast, are not tied to such discretizations, which allows them to scale more naturally to large action spaces. Moreover, our use of the generalized-exponential family (Section 3.1) can be understood as a form of *regularization*, stabilizing learning dynamics and reducing variance, as seen in the sensitivity analysis of Figure 3. As a result, policy gradient approaches provide both flexibility and robustness, adapting effectively to workload variations in large-scale queue-aware scheduling.

## 5 RELATED WORK

Our work is at the intersection of two lines of research: (i) robust policy optimization methods based on generalized exponential families and (ii) systems approaches for efficient edge inference.

Traditional policy optimization methods often rely on Gaussian policies due to their tractability in continuous action spaces. Recent work by Zhu et al. (2025) introduced the $\delta$-*exponential family* as a broader class of policies that generalizes beyond the Gaussian, allowing for both light-tailed and heavy-tailed behaviors. Their results show that heavy-tailed policies, such as the $\delta$-Gaussian or Student's $t$, can improve robustness and stability in reinforcement learning. Our use of $\delta$-softmax (Tsallis-softmax) in queue-aware threshold selection builds directly on these insights: we adapt the same $\delta$-exponential machinery to discrete thresholds, giving us a tunable way to balance exploration and exploitation in stochastic queuing environments.

On the systems side, several works have explored how to optimize inference under resource constraints by selectively allocating computation across edge and cloud resources. Early frameworks such as Neurosurgeon (Kang et al., 2017), Edge AI (Li et al., 2020), and distributed inference over heterogeneous devices (Hu & Li, 2022) established the benefits of adaptive inference offloading. More recent work has focused on hierarchical inference, including module selection at the edge (Behera et al., 2023), algorithms for balancing accuracy and delay (Moothedath et al., 2024; Eytur et al., 2024), regret bounds for online learning (Al-Atat et al., 2024), and offloading algorithms for accuracy maximization (Fresa & Champati, 2023). Other proposals emphasize application-specific scenarios such as live video analytics for drones (Wang et al., 2018) and mobile inference balancing latency and accuracy (Ogden & Guo, 2020). Our work is complementary: instead of focusing on which device should process an input, we study *when to stop* within an early-exit network, making the exit decision queue-aware and adaptive to system congestion.

Closer to our setting, prior work on scheduling inputs in early-exit neural networks (Casale & Roveri, 2023) estimates exit policies under a simplifying assumption that the final-layer confidence $C_L$ is independent of the threshold $\alpha$. This assumption poses potential issues: in practice, $C_L$ depends on $\alpha$, since early exits filter out easy inputs and shift the distribution of samples reaching deeper layers. In contrast, our bandit-based formulation avoids this issue by directly using observed $(C_I, C_L)$ pairs to define rewards, thereby adapting online to how threshold decisions shape the inputs that reach deeper layers. Similarly, CEED (Chen et al., 2025) requires training multiple neural predictors, whereas our lightweight, decentralized controller can leverage recent advances in robust bandit theory (Tsallis-softmax (Zhu et al., 2025)) to efficiently schedule early exits in real time.

In summary, prior work on the $\delta$-exponential family provides the theoretical foundation for robust exploration, and selective query frameworks show the systems value of adaptive inference. Our work combines these two perspectives by integrating generalized policy optimization with buffer-aware scheduling for early-exit neural networks, validated through real testbed experiments.

## 6 CONCLUSION

This paper reframed early exits as a *systems scheduling* problem and validated the approach on a heterogeneous edge testbed. We showed that buffer-aware thresholds materially improve performance, that $\delta$-softmax (Tsallis-softmax) policies stabilize learning under stochastic loads, and that monotone parametric thresholds exploit domain structure for efficient generalization. More broadly, our findings underscore the power of the generalized exponential family: by shaping exploration and robustness, it provides a principled foundation for designing adaptive policies that respond to system dynamics.

This work opens up a number of avenues for further research, including extensions to multi-exit architectures and adversarial workloads. More broadly, incorporating energy and fairness metrics into this framework can position buffer-aware EENNs as a foundation for sustainable edge intelligence. Notably, much of the recent work on hierarchical inference and the edge–cloud continuum can be cast as variations of the same fundamental question we address here: *when to early exit*.

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

## A  NOTATION AND METRICS

Table 2 summarizes the acronyms of the algorithms considered in this work, together with their parameters.

Table 2: Experimental configurations: algorithms and parameters.

|  | Algorithm | Parameters / Description |
|---|---|---|
| Baseline | Baseline | Original scheduler, no buffer awareness |
| UCB | UCB (Vanilla) | Buffer-aware scheduler (fixed thresholds) |
| UCB-Mono | UCB (Robust, Monotonic) | Linear thresholds $\alpha(q) = \theta_1 q + \theta_2, \theta_1 \leq 0$ |
| PG | Policy Gradient (Vanilla) | Standard softmax ($\delta = 1$) |
| TSA-PG($\gamma$) | Policy Gradient (Robust) | Tsallis-softmax with $\delta = \gamma$ |

Table 3 summarizes notation used throughout this work.

Table 3: Notation used throughout the paper. Vectors are bold; scalars are plain.

| Symbol | Description |
|---|---|
| $\mathbf{x}$ | Input (image/example) |
| $\mathbf{z}_I$, $\mathbf{z}_L$ | Logits at intermediate side branch and final layer |
| $\mathbf{p}_I \triangleq \mathrm{softmax}(\mathbf{z}_I), \mathbf{p}_L \triangleq \mathrm{softmax}(\mathbf{z}_L)$ | Class probability vectors at intermediate/final layers |
| $C_I \triangleq \max_c [\mathbf{p}_I]_c, C_L \triangleq \max_c [\mathbf{p}_L]_c$ | Confidence (max class prob.) at intermediate/final layers |
| $\hat{y}$ | Predicted class label (early or final) |
| $\alpha$ | Confidence threshold for early exit |
| $\mathcal{A}$ | Discrete action set of candidate thresholds |
| $\Delta C \triangleq \max\{C_L - C_I, 0\}$ | Confidence gain from continuing past the side branch |
| $o, o(q)$ | Overhead (fixed) or queue-aware overhead (e.g., $o(q) = \nu q - \kappa$) |
| $\mu, \kappa$ | Overhead parameters in $o(q)$ |
| $B$ | Buffer capacity |
| $q = |Q|$ | Backlog (current number of queued inputs) |
| $r_t(\alpha_t)$ | Instantaneous reward at round $t$ (equation 1) |
| $t, T$ | Round index and time horizon |
| $N_t(\alpha)$ | Times threshold $\alpha$ was selected up to $t$ |
| $Q_t(\alpha), \widehat{Q}(\cdot)$ | Empirical mean reward estimate (tabular/UCB) |
| $\beta$ | UCB exploration parameter |
| $\mathcal{H}$ | Policy class for contextual UCB (e.g., monotone linear) |
| $h_\theta(q)$ | Parametric threshold policy $\mathrm{clip}_{[0,1]}(\theta_1 q + \theta_2)$ |
| $\theta, \theta_q(\alpha)$ | Policy parameters (global or per-backlog, PG methods) |
| $\pi^{(q)}(\alpha)$ | Stochastic policy over thresholds given backlog $q$ |
| $\eta$ | Learning rate (policy gradient) |
| $B_t$ | Baseline for variance reduction (policy gradient) |
| $\delta$ | Tsallis / $\delta$-exponential shape parameter ($\delta=1$ is softmax) |
| $\exp_\delta(\cdot), \log_\delta(\cdot)$ | Generalized (Tsallis) exponential / logarithm |
| $Q_{\max}$ | Maximum queue size tracked by the learner |

### A.1  EVALUATION METRICS

This appendix formalizes the metrics reported in the work. Let $N_{\mathrm{tot}}$ denote the total number of arrivals to the system, $N_{\mathrm{proc}}$ the number of processed samples, $N_{\mathrm{corr}}$ the number of correctly classified samples, and $N_{\mathrm{lost}}$ the number of lost samples. We let $T$ denote the total experiment time.

**Workload.**   Except otherwise noted, we assume that arrivals occur according to a Poisson process with rate $\lambda$,

$$\lambda = \frac{N_{\mathrm{tot}}}{T}.$$

**Worker.** A worker represents the execution resource (e.g., edge or cloud). If $\mathcal{W} = \{\text{edge}, \text{cloud}\}$, then

$$N_{\text{proc}} = N_{\text{edge}} + N_{\text{cloud}}.$$

**Loss rate and loss ratio.** The instantaneous loss rate is

$$r_{\text{loss}}(t) = \frac{N_{\text{lost}}(t)}{t},$$

and the average loss rate over the whole simulation is

$$r_{\text{loss}} = \frac{N_{\text{lost}}}{T}.$$

The loss ratio is the fraction of arrivals that were lost due to buffer overflow,

$$\text{LossRatio} = \frac{N_{\text{lost}}}{N_{\text{tot}}}.$$

**Throughput.** Throughput is the rate of processed samples per unit of time

$$\Theta = \frac{N_{\text{proc}}}{T}.$$

**Utilization.** Utilization measures the fraction of time the server is busy

$$U = \frac{\mathcal{B}}{T},$$

where $\mathcal{B}$ is the total busy time.

**Accuracy.** The accuracy is given by

$$\text{Accuracy} = \frac{N_{\text{corr}}}{N_{\text{proc}}}.$$

**Score.** The score combines accuracy and loss into a single metric.

$$\text{Score} = \text{Accuracy} - \text{LossRatio}.$$

We also define the goodput as

$$\text{Goodput} = \frac{N_{\text{corr}}}{T} = \text{Accuracy} \cdot (1 - \text{LossRatio}) \cdot \lambda = \text{Accuracy} \cdot (\lambda - r_{\text{loss}}).$$

Higher score and goodput values indicate policies that jointly maximize accuracy while minimizing loss.

# B BACKGROUND

At the inference phase, the EENN model processes an input $\mathbf{x}$ layer by layer on the edge device until it reaches an intermediate exit branch, which produces a logit vector $\mathbf{z}_I(\mathbf{x})$. The exit branch then generates a probability vector $\mathbf{p}_I(\mathbf{x}) \triangleq \mathrm{softmax}(\mathbf{z}_I(\mathbf{x}))$. Each component $[p_I(\mathbf{x})]_c$ of $\mathbf{p}_I(\mathbf{x})$ represents the probability of $\mathbf{x}$ belonging to class $c \in C$, where $C$ is the set of possible classes. The classification confidence estimate is defined as $C_I(\mathbf{x}) \triangleq \max_c[\mathbf{p}_I(\mathbf{x})]_c$. If $C_I(\mathbf{x}) \geq \alpha$, where $\alpha \in [0, 1]$ is the confidence threshold, the input $\mathbf{x}$ is classified locally as $\hat{y}_I(\mathbf{x}) = \arg\max_{c \in C}[\mathbf{p}_I(\mathbf{x})]_c$. Otherwise, when $C_I(\mathbf{x}) < \alpha$, the classification is deemed unreliable, and the edge device offloads the data to the cloud, incurring an overhead $o$. The cloud processes the remaining layers until the final layer is reached, producing the final probability vector $\mathbf{p}_L(\mathbf{x})$, the confidence estimate, $C_L(\mathbf{x}) \triangleq \max_c[\mathbf{p}_L(\mathbf{x})]_c$, and the final prediction $\hat{y}_L(\mathbf{x}) = \arg\max_{c \in C}[\mathbf{p}_L(\mathbf{x})]_c$. Henceforth, we drop the explicit dependence of $C_I(\mathbf{x})$, $C_L(\mathbf{x})$, and all variables derived from them on $\mathbf{x}$ whenever the input is clear from context.

# C REGRET ANALYSIS FOR CONTEXTUAL UCB WITH DISCRETIZED QUEUE

We provide a finite-time regret guaranty for a simplified variant of our UCB-based early-exit policy, formulated as a stochastic contextual bandit with a finite context space. This setting corresponds to a discretized queue length and stationary reward distributions and is used as a baseline theoretical model; the full queuing dynamics considered in the main paper are more general and may induce non-stationarity.

## C.1 MODEL

We consider a stochastic contextual bandit problem with the following components:

- A finite set of contexts $\mathcal{Q} = \{0, 1, \ldots, Q_{\max}\}$, representing a discretized queue length.

- A finite set of actions (thresholds) $\mathcal{A} = \{1, \ldots, K\}$.

- At each round $t = 1, \ldots, T$, a context $q_t \in \mathcal{Q}$ is observed, an action $\alpha_t \in \mathcal{A}$ is chosen, and a reward $r_t \in [0, 1]$ is observed.

- For each pair $(q, \alpha) \in \mathcal{Q} \times \mathcal{A}$, there is an unknown distribution $\nu_{q,\alpha}$ with mean

$$\mu(q, \alpha) := \mathbb{E}_{r \sim \nu_{q,\alpha}}[r],$$

  and we assume that all rewards are $\sigma$-sub-Gaussian (which holds in our setting, since all our rewards are limited in $[0, 1]$). Conditioned on given $(q_t, \alpha_t)$, the rewards are i.i.d. draws from $\nu_{q_t, \alpha_t}$.

We assume that the sequence of contexts $(q_t)_{t=1}^T$ is exogenous (independent of the learner's actions) and arbitrary; for example, it may be i.i.d. from an unknown distribution over $\mathcal{Q}$. For each context $q$, we define an optimal action

$$\alpha^\star(q) \in \arg\max_{\alpha \in \mathcal{A}} \mu(q, \alpha),$$

and gaps

$$\Delta(q, \alpha) := \mu(q, \alpha^\star(q)) - \mu(q, \alpha) \geq 0, \quad \alpha \in \mathcal{A}.$$

We assume $\Delta(q, \alpha) > 0$ for all suboptimal actions $\alpha \neq \alpha^\star(q)$ and all $q$.

The (pseudo-)regret after $T$ rounds is defined as

$$R_T := \sum_{t=1}^T \left( \mu(q_t, \alpha^\star(q_t)) - \mu(q_t, \alpha_t) \right) = \sum_{q \in \mathcal{Q}} \sum_{\alpha \in \mathcal{A}} \Delta(q, \alpha) \, \mathbb{E}\big[N_T(q, \alpha)\big], \tag{7}$$

where $N_T(q, \alpha)$ is the number of times action $\alpha$ is selected under context $q$ up to time $T$.

## C.2 CONTEXTUAL UCB ALGORITHM

We consider the following contextual UCB strategy, which applies a standard UCB rule independently for each context $q$.

For each $(q, \alpha) \in \mathcal{Q} \times \mathcal{A}$ and each round $t$, let

$$N_t(q, \alpha) := \sum_{s=1}^{t-1} \mathbf{1}\{q_s = q, \alpha_s = \alpha\}$$

denote the number of times action $\alpha$ has been selected when the context was $q$ up to time $t - 1$, and let

$$\hat{\mu}_t(q, \alpha) := \begin{cases} \dfrac{1}{N_t(q, \alpha)} \sum_{s=1}^{t-1} r_s \mathbf{1}\{q_s = q, \alpha_s = \alpha\}, & \text{if } N_t(q, \alpha) > 0, \\ 0, & \text{if } N_t(q, \alpha) = 0. \end{cases}$$

Given a time horizon $T$, we define the UCB index for context $q_t$ and action $\alpha$ as

$$U_t(q_t, \alpha) := \begin{cases} \hat{\mu}_t(q_t, \alpha) + \beta \sqrt{\dfrac{\log T}{N_t(q_t, \alpha)}}, & \text{if } N_t(q_t, \alpha) > 0, \\ +\infty, & \text{if } N_t(q_t, \alpha) = 0. \end{cases}$$

At each round $t$, after observing $q_t$, the algorithm selects

$$\alpha_t \in \arg \max_{\alpha \in \mathcal{A}} U_t(q_t, \alpha).$$

This is exactly UCB (Auer et al., 2002), with $\beta = \sqrt{2}$, applied separately to each context $q$, using the same confidence parameter $\sqrt{(2 \log T)/N}$. In (4), the term in the denominator is $N_t + 1$ instead of $N_t$, which corresponds to pulling each arm once, as initialization.

## C.3 REGRET BOUND

We now state the regret guarantee.

**Theorem C.1 (Regret of contextual UCB with finite context space)** *Under the model and assumptions above, the expected regret of the contextual UCB algorithm satisfies*

$$R_T \leq C \sum_{q \in \mathcal{Q}} \sum_{\alpha \neq \alpha^\star(q)} \frac{\log T}{\Delta(q, \alpha)} = \tilde{O}(Q_{\max} K \log T), \tag{8}$$

*for some universal constant $C > 0$. In particular, the regret grows at most logarithmically with the horizon $T$ and linearly with the number of contexts $|\mathcal{Q}| = Q_{\max} + 1$ and actions $K$.*

Fix a context $q \in \mathcal{Q}$ and a suboptimal action $\alpha \neq \alpha^\star(q)$ with gap $\Delta(q, \alpha) > 0$. Let $T_q$ denote the (random) number of rounds up to time $T$ in which the context $q_t$ equals $q$:

$$T_q := \sum_{t=1}^{T} \mathbf{1}\{q_t = q\}.$$

Conditioned on the context sequence $(q_t)$, the rounds for which $q_t = q$ define a standard $K$-armed stochastic bandit problem with horizon $T_q$, where arm $\alpha$ has mean reward $\mu(q, \alpha)$ and the algorithm applies exactly UCB on that subsequence (because the UCB indices and choices for context $q$ depend only on rewards observed when $q_t = q$).

By the standard UCB analysis (Auer et al., 2002), for each fixed $q$ and $\alpha \neq \alpha^\star(q)$ we have

$$\mathbb{E}\big[N_T(q, \alpha) \,\big|\, (q_t)_{t=1}^{T}\big] \leq \frac{8 \log T}{\Delta(q, \alpha)^2} + 1 + \frac{\pi^2}{3}. \tag{9}$$

The bound uses $\log T$ instead of $\log T_q$; since $T_q \leq T$, this only makes the bound looser and is thus valid.

Taking expectations with respect to the context sequence and using the tower property, we obtain

$$\mathbb{E}\big[N_T(q,\alpha)\big] \leq \frac{8\log T}{\Delta(q,\alpha)^2} + 1 + \frac{\pi^2}{3}.$$

Substituting this into the regret decomposition equation 7, we get

$$
\begin{aligned}
R_T &= \sum_{q\in\mathcal{Q}}\sum_{\alpha\in\mathcal{A}} \Delta(q,\alpha)\,\mathbb{E}\big[N_T(q,\alpha)\big] \\
&= \sum_{q\in\mathcal{Q}}\sum_{\alpha\neq\alpha^\star(q)} \Delta(q,\alpha)\,\mathbb{E}\big[N_T(q,\alpha)\big] \\
&\leq \sum_{q\in\mathcal{Q}}\sum_{\alpha\neq\alpha^\star(q)} \Delta(q,\alpha)\left(\frac{8\log T}{\Delta(q,\alpha)^2} + 1 + \frac{\pi^2}{3}\right) \\
&\leq C \sum_{q\in\mathcal{Q}}\sum_{\alpha\neq\alpha^\star(q)} \frac{\log T}{\Delta(q,\alpha)},
\end{aligned}
$$

for some universal constant $C > 0$ (e.g., $C$ can absorb the additive constants and the factors $8/\Delta(q,\alpha)$, using that $\Delta(q,\alpha) \leq 1$ because rewards lie in $[0,1]$). This proves equation 8 and the claimed $\tilde{O}(Q_{\max}K\log T)$ scaling.

Theorem C.1 should be interpreted as an illustrative regret guarantee for a *simplified* version of our problem, in which the queue length is discretized, and the context sequence $(q_t)$ is treated as exogenous and independent of the learner's actions. In this scenario, our monotone-UCB policy in the main text restricts the policy class to monotone thresholds $\alpha(q)$, which reduces the number of effective degrees of freedom. In this finite-context, stationary setting, contextual UCB reduces to running standard UCB per context and thus enjoys the usual logarithmic regret guarantees. In our actual early-exit problem, however, the queue length $q_t$ is *endogenous*: it evolves according to the queueing dynamics and is directly affected by the chosen thresholds (actions), so that both the context process and the reward distributions become action-dependent and potentially non-stationary. A full regret analysis in this stateful, queue-dependent setting would require a more elaborate treatment (closer to Markovian bandits or simple MDPs with queueing dynamics), which is beyond the scope of this paper. We therefore present Theorem C.1 as a first, illustrative regret result that clarifies how our UCB-based policy fits into the classical stochastic contextual bandit framework under a discretized, exogenous-queue approximation.

## D  Q-EXPONENTIAL CALCULUS AND THE $\delta$-SOFTMAX POLICY GRADIENT

This appendix contains the definitions and identities underlying the $\delta$-softmax policy in the main text, and derives the policy-gradient update used in Algorithm 1.

Despite the growing interest in the $\delta$-exponential family (e.g., (Zhu et al., 2025)), to the best of our knowledge the gradient-ascent rule for $\delta$-softmax (Tsallis-softmax) policies is novel to this work.

### D.1  GENERALIZED LOGARITHM AND EXPONENTIAL

For $\delta \in \mathbb{R}$, define the generalized (Tsallis) logarithm and exponential:

$$
\log_\delta(u) = \begin{cases} \dfrac{u^{1-\delta}-1}{1-\delta}, & \delta \neq 1, \\ \log u, & \delta = 1, \end{cases}
\qquad
\exp_\delta(u) = \begin{cases} \big(1+(1-\delta)u\big)^{\frac{1}{1-\delta}}, & \delta \neq 1, \\ \exp(u), & \delta = 1, \end{cases}
$$

with inverse relations $\log_\delta(\exp_\delta u) = u$ and $\exp_\delta(\log_\delta v) = v$ when $u$ and $v$ are in range. Two useful identities are:

$$\log_\delta(uv) = \log_\delta u + \log_\delta v + (1-\delta)\log_\delta u \log_\delta v, \tag{10}$$

$$\frac{\partial}{\partial u}\log_\delta(u) = u^{-\delta}. \tag{11}$$

## D.2 THE $\delta$-SOFTMAX POLICY OVER DISCRETE ACTIONS

Let $\mathcal{A}$ be the finite set of thresholds. For parameters $\theta(\alpha) \in \mathbb{R}$, define

$$\pi(\alpha) = \frac{\exp_\delta\big(\theta(\alpha)\big)}{\sum_{b \in \mathcal{A}} \exp_\delta\big(\theta(b)\big)}.$$

When $\delta = 1$ this reduces to the standard softmax. When $\delta \neq 1$, the family produces heavier- or lighter-tailed discrete distributions depending on $\delta$, enabling tunable exploration.

## D.3 EXPECTED REWARD AND BASELINE

Let $\mu(\alpha) = \mathbb{E}[r \mid \alpha]$ denote the per-action value and $B$ any baseline independent of the sampled action. The expected reward is

$$J(\theta) = \mathbb{E}_{\alpha \sim \pi}[\mu(\alpha)] = \sum_{\alpha \in \mathcal{A}} \mu(\alpha)\,\pi(\alpha).$$

Subtracting $B$ does not change the gradient because $\sum_\alpha \pi(\alpha) = 1$.

## D.4 GRADIENT OF $J(\theta)$ UNDER $\delta$-SOFTMAX

Write $S = \sum_b \exp_\delta(\theta(b))$ and $f_a = \exp_\delta(\theta(a))$ for brevity, so $\pi(a) = f_a/S$. Differentiating $\pi(b)$ with respect to $\theta(a)$ gives

$$\frac{\partial \pi(b)}{\partial \theta(a)} = \frac{1}{S}\frac{\partial f_b}{\partial \theta(a)} - \frac{f_b}{S^2}\frac{\partial S}{\partial \theta(a)}.$$

Using $\dfrac{\partial}{\partial u}\exp_\delta(u) = \exp_\delta(u)^\delta$ (the inverse of equation 11), we obtain

$$\frac{\partial f_b}{\partial \theta(a)} = \delta_{ab}\,f_b^\delta, \qquad \frac{\partial S}{\partial \theta(a)} = f_a^\delta,$$

where $\delta_{ab}$ is Kronecker's delta. Hence

$$\frac{\partial \pi(b)}{\partial \theta(a)} = \frac{\delta_{ab}\,f_b^\delta}{S} - \frac{f_b}{S}\frac{f_a^\delta}{S} = \delta_{ab}\frac{f_b^\delta}{S} - \pi(b)\frac{f_a^\delta}{S}.$$

Noting that $f_x^\delta = \big(\exp_\delta(\theta(x))\big)^\delta$ and the algebraic identity

$$\frac{f_x^\delta}{S} = \frac{\pi(x)}{1 + (1-\delta)\theta(x)},$$

(which follows by differentiating $\log_\delta S$ via equation 10 and equation 11), we can write

$$\frac{\partial \pi(b)}{\partial \theta(a)} = \delta_{ab}\frac{\pi(b)}{1 + (1-\delta)\theta(b)} - \pi(b)\frac{\pi(a)}{1 + (1-\delta)\theta(a)}.$$

Therefore,

$$\frac{\partial J}{\partial \theta(a)} = \sum_b \mu(b)\frac{\partial \pi(b)}{\partial \theta(a)}$$

$$= \mathbb{E}_{b \sim \pi}\left[\big(\mu(b) - B\big)\left(\frac{\mathbb{1}\{b = a\}}{1 + (1-\delta)\theta(b)} - \frac{\pi(a)}{1 + (1-\delta)\theta(a)}\right)\right], \qquad (12)$$

where the baseline term $B$ vanishes upon summation. In the limit $\delta \to 1$,

$$\frac{1}{1 + (1-\delta)\theta(\cdot)} \to 1 \quad \Rightarrow \quad \frac{\partial J}{\partial \theta(a)} = \mathbb{E}_{b \sim \pi}\big[(\mu(b) - B)\big(\mathbb{1}\{b = a\} - \pi(a)\big)\big],$$

recovering the standard softmax policy-gradient.

### D.5 MAPPING TO ALGORITHM 1

In the main text, parameters are indexed by backlog $q$: $\theta_q(\alpha)$, the baseline is $B_t$, and the per-round reward is

$$
r_t = \begin{cases} 0, & C_I \geq \alpha_t, \\ \max(C_L - C_I, 0) - (\mu q - \kappa), & \text{otherwise.} \end{cases}
$$

Stochastic gradient ascent with step size $\eta$ applies equation 12 to the sampled $(q, \alpha_t)$, yielding the update (for every $\alpha \in \mathcal{A}$)

$$
\theta_q(\alpha) \leftarrow \theta_q(\alpha) + \eta\,(r_t - B_t) \times \begin{cases} \mathbb{1}\{\alpha = \alpha_t\} - \pi^{(q)}(\alpha), & \delta = 1, \\[2mm] \dfrac{\mathbb{1}\{\alpha = \alpha_t\} - \pi^{(q)}(\alpha)}{1 + (1-\delta)\theta_q(\alpha)}, & \delta \neq 1, \end{cases}
$$

which is exactly the rule implemented in Algorithm 1.

Our Tsallis policy gradient can be interpreted as applying a Box–Cox-type transform (Box & Cox, 1964) to policy probabilities/advantages, compressing extreme values and empirically stabilizing gradients. From the RL side, our update fits into the Tsallis-entropy-regularized RL literature and sparse policies, where Tsallis regularization yields sparse, more stable policies than Shannon-entropy softmax. We now emphasize that there exists a rich body of work establishing non-asymptotic convergence rates for *classical* policy gradient under structural assumptions (e.g., LQR and tabular/linear MDPs (Fazel et al., 2018; Agarwal et al., 2021; Wang et al., 2020)), but our queue-dependent early-exit setting, with endogenous non-stationary contexts, falls outside the scope of these analyses. We therefore present our Tsallis / $\delta$-softmax PG as a principled, Box–Cox/Tsallis-inspired design with empirical robustness, and explicitly leave a full convergence-rate theory in this setting to future work.

### D.6 IMPLEMENTATION NOTES

- **Domain guard.** Ensure $1 + (1 - \delta)\theta_q(\alpha) > 0$ for all $(q, \alpha)$ to keep $\exp_\delta$ real-valued. A practical choice is to reparameterize $\theta_q(\alpha) = \frac{z_q(\alpha)}{1-\delta} - \epsilon$ with $z_q(\alpha) > 0$, or to clip to a small positive margin.

- **Baselines.** Any $B_t$ independent of the sampled action is unbiased (e.g., running mean of recent rewards or a value-function estimate) and reduces variance.

- **Softmax branch.** Handle $\delta = 1$ explicitly for numerical stability; do not attempt to approximate it via limits in code.

- **Temperature.** An optional temperature $\tau > 0$ can be introduced by replacing $\theta \mapsto \theta/\tau$, orthogonal to $\delta$; $\tau$ scales logits, while $\delta$ shapes tails.

The implementation of our $\delta$-softmax policy gradient follows the above design notes. We highlight how each element is realized in practice:

- **Domain guard.** In code, the domain restriction $1 + (1 - \delta)\theta_q(\alpha) > 0$ is enforced by clipping. Specifically, the $\delta$-exponential is implemented as `np.maximum(1 + (1 - delta) * u, 1e-8) ** (1 / (1 - delta))`, ensuring the base remains strictly positive and thus keeping $\exp_\delta$ real-valued.

- **Baselines.** To reduce variance, we maintain an exponential moving average baseline: `baseline = beta * baseline + (1 - beta) * reward`. The adjusted reward `reward - baseline` is then used in the update step, which is unbiased since the baseline is independent of the sampled action.

- **Softmax branch.** For the special case $\delta = 1$, the implementation switches explicitly to the standard exponential: `if delta == 1: return np.exp(u)`. This avoids numerical instability and removes the need to approximate the $\delta \to 1$ limit.

- **Temperature.** While the design allows for an optional temperature parameter $\tau > 0$ (applied via $\theta \mapsto \theta/\tau$), our current implementation does not include $\tau$. This extension is straightforward and orthogonal to $\delta$, scaling logits without changing the tail behavior.

# E  PER-MACHINE METRICS

This appendix reports detailed *per-machine* performance metrics, broken down by worker type (Raspberry Pi and MiniPC) and workload scenario. While the main text emphasizes aggregate system-level behavior, these tables highlight how individual hardware units contribute to overall performance. They provide additional insight into resource utilization and the trade-offs achieved by different policies.

Several general patterns emerge:

- **Arrival rates.** The effective arrival rate observed at each worker reflects both exogenous traffic and the impact of early exits. The Raspberry Pi often operates near its capacity.

- **Loss behavior.** The Raspberry Pi experiences significant loss under baseline policies, whereas advanced methods (UCB, UCB-Mono, TSA-PG) reduce the loss rate and loss ratio markedly. The MiniPC rarely drops samples, underscoring its role as a stable back-end.

- **Throughput and utilization.** Despite its limited capacity, the Raspberry Pi achieves high utilization ($> 95\%$) under most methods, showing that the system effectively exploits edge resources. The MiniPC, by contrast, operates at low utilization.

- **Policy effects.** UCB-based and TSA-PG policies consistently improve throughput on the Raspberry Pi while lowering losses, suggesting that intelligent threshold adaptation is essential to balance early exits and full processing.

Comparing across scenarios, we observe systematic differences:

- In the **Stationary** scenario, performance is stable and policies mainly affect loss reduction, with TSA-PG achieving the best trade-off.

- In the **Easy→Hard** scenario, losses on the Raspberry Pi increase due to progressively more difficult inputs, but adaptive methods sustain throughput by shifting part of the load to the MiniPC.

- In the **Hard→Easy** scenario, initial congestion is more severe than in Easy→Hard, since early difficult samples saturate the Raspberry Pi; however, policies that adapt quickly still recover performance as the workload eases.

Overall, these per-machine measurements confirm that the performance gains reported in the main text stem not only from aggregate improvements but also from better balancing of heterogeneous resources.

## E.1 STATIONARY SCENARIO

Table 4: Per-machine metrics (stationary scenario).

| Method | Worker | Arrival rate | Loss rate | Loss ratio | Throughput | Utilization |
|---|---|---|---|---|---|---|
| Baseline | Raspberry Pi | $3.9697 \pm 0.0238$ | $0.7865 \pm 0.0165$ | $0.1981 \pm 0.0031$ | $3.1831 \pm 0.0064$ | $0.9976 \pm 0.0007$ |
| | MiniPC | $2.1537 \pm 0.0050$ | $0.0000 \pm 0.0000$ | $0.0000 \pm 0.0000$ | $2.1537 \pm 0.0050$ | $0.0822 \pm 0.0002$ |
| UCB | Raspberry Pi | $3.9720 \pm 0.0222$ | $0.3583 \pm 0.0139$ | $0.0902 \pm 0.0030$ | $3.6138 \pm 0.0110$ | $0.9844 \pm 0.0015$ |
| | MiniPC | $2.1489 \pm 0.0042$ | $0.0000 \pm 0.0000$ | $0.0000 \pm 0.0000$ | $1.8307 \pm 0.0059$ | $0.0655 \pm 0.0003$ |
| UCB-Mono | Raspberry Pi | $3.9741 \pm 0.0198$ | $0.2902 \pm 0.0132$ | $0.0730 \pm 0.0030$ | $3.6840 \pm 0.0110$ | $0.9567 \pm 0.0025$ |
| | MiniPC | $2.1667 \pm 0.0136$ | $0.0000 \pm 0.0000$ | $0.0000 \pm 0.0000$ | $2.1667 \pm 0.0136$ | $0.0714 \pm 0.0004$ |
| TSA-PG(0.50) | Raspberry Pi | $3.987 \pm 0.0165$ | $0.2143 \pm 0.0092$ | $0.0539 \pm 0.0021$ | $3.7598 \pm 0.0129$ | $0.9646 \pm 0.0021$ |
| | MiniPC | $2.0776 \pm 0.0081$ | $0.0000 \pm 0.0000$ | $0.0000 \pm 0.0000$ | $2.0776 \pm 0.0081$ | $0.0685 \pm 0.0003$ |
| TSA-PG(0.75) | Raspberry Pi | $3.9774 \pm 0.0198$ | $0.2140 \pm 0.0095$ | $0.0539 \pm 0.0022$ | $3.7601 \pm 0.0130$ | $0.9645 \pm 0.0019$ |
| | MiniPC | $2.0767 \pm 0.0062$ | $0.0000 \pm 0.0000$ | $0.0000 \pm 0.0000$ | $2.0767 \pm 0.0062$ | $0.0685 \pm 0.0003$ |
| TSA-PG(1.00) | Raspberry Pi | $3.9826 \pm 0.0124$ | $0.2053 \pm 0.0085$ | $0.0517 \pm 0.0020$ | $3.7688 \pm 0.0147$ | $0.9645 \pm 0.0018$ |
| | MiniPC | $2.0622 \pm 0.0098$ | $0.0000 \pm 0.0000$ | $0.0000 \pm 0.0000$ | $2.0622 \pm 0.0098$ | $0.0680 \pm 0.0004$ |
| TSA-PG(1.25) | Raspberry Pi | $3.9457 \pm 0.0204$ | $0.1748 \pm 0.0078$ | $0.0440 \pm 0.0018$ | $3.7993 \pm 0.0160$ | $0.9643 \pm 0.0016$ |
| | MiniPC | $2.0087 \pm 0.0147$ | $0.0000 \pm 0.0000$ | $0.0000 \pm 0.0000$ | $2.0087 \pm 0.0147$ | $0.0662 \pm 0.0005$ |
| TSA-PG(1.50) | Raspberry Pi | $3.9812 \pm 0.0145$ | $0.1596 \pm 0.0084$ | $0.0401 \pm 0.0020$ | $3.8146 \pm 0.0158$ | $0.9652 \pm 0.0024$ |
| | MiniPC | $1.9905 \pm 0.0195$ | $0.0000 \pm 0.0000$ | $0.0000 \pm 0.0000$ | $1.9905 \pm 0.0195$ | $0.0656 \pm 0.0006$ |
| TSA-PG(1.75) | Raspberry Pi | $3.9741 \pm 0.0136$ | $0.1488 \pm 0.0093$ | $0.0374 \pm 0.0023$ | $3.8253 \pm 0.0171$ | $0.9658 \pm 0.0022$ |
| | MiniPC | $1.9769 \pm 0.0244$ | $0.0000 \pm 0.0000$ | $0.0000 \pm 0.0000$ | $1.9769 \pm 0.0244$ | $0.0652 \pm 0.0008$ |
| TSA-PG(2.00) | Raspberry Pi | $3.9874 \pm 0.0191$ | $0.1535 \pm 0.0105$ | $0.0386 \pm 0.0026$ | $3.8207 \pm 0.0193$ | $0.9655 \pm 0.0025$ |
| | MiniPC | $1.9818 \pm 0.0285$ | $0.0000 \pm 0.0000$ | $0.0000 \pm 0.0000$ | $1.9818 \pm 0.0285$ | $0.0653 \pm 0.0010$ |
| TSA-PG(2.50) | Raspberry Pi | $3.9901 \pm 0.0200$ | $0.1806 \pm 0.0124$ | $0.0443 \pm 0.0030$ | $3.8925 \pm 0.0222$ | $0.9717 \pm 0.0023$ |
| | MiniPC | $1.9505 \pm 0.0298$ | $0.0000 \pm 0.0000$ | $0.0000 \pm 0.0000$ | $1.9505 \pm 0.0298$ | $0.0641 \pm 0.0010$ |

## E.2 EASY→HARD SCENARIO

Table 5: Per-machine metrics (easy→hard scenario).

| Method | Worker | Arrival rate | Loss rate | Loss ratio | Throughput | Utilization |
|---|---|---|---|---|---|---|
| UCB | Raspberry Pi | $3.9731 \pm 0.0182$ | $0.5368 \pm 0.0146$ | $0.1351 \pm 0.0030$ | $3.4352 \pm 0.0105$ | $0.9913 \pm 0.0009$ |
|  | MiniPC | $2.1889 \pm 0.0058$ | $0.0000 \pm 0.0000$ | $0.0000 \pm 0.0000$ | $2.1889 \pm 0.0058$ | $0.0782 \pm 0.0002$ |
| Baseline | Raspberry Pi | $3.9767 \pm 0.0282$ | $0.8955 \pm 0.0169$ | $0.2256 \pm 0.0030$ | $3.0741 \pm 0.0063$ | $0.9983 \pm 0.0004$ |
|  | MiniPC | $2.3447 \pm 0.0055$ | $0.0000 \pm 0.0000$ | $0.0000 \pm 0.0000$ | $2.3447 \pm 0.0055$ | $0.0895 \pm 0.0002$ |
| UCB-Mono | Raspberry Pi | $3.9851 \pm 0.0291$ | $0.4391 \pm 0.0147$ | $0.1105 \pm 0.0032$ | $3.5350 \pm 0.0088$ | $0.9738 \pm 0.0018$ |
|  | MiniPC | $2.5673 \pm 0.0094$ | $0.0000 \pm 0.0000$ | $0.0000 \pm 0.0000$ | $2.5673 \pm 0.0094$ | $0.0846 \pm 0.0003$ |
| TSA-PG(0.50) | Raspberry Pi | $3.9898 \pm 0.0228$ | $0.3484 \pm 0.0118$ | $0.0877 \pm 0.0026$ | $3.6258 \pm 0.0110$ | $0.9781 \pm 0.0016$ |
|  | MiniPC | $2.4224 \pm 0.0061$ | $0.0000 \pm 0.0000$ | $0.0000 \pm 0.0000$ | $2.4224 \pm 0.0061$ | $0.0799 \pm 0.0002$ |
| TSA-PG(0.75) | Raspberry Pi | $3.9699 \pm 0.0188$ | $0.3458 \pm 0.0116$ | $0.0870 \pm 0.0025$ | $3.6283 \pm 0.0106$ | $0.9779 \pm 0.0015$ |
|  | MiniPC | $2.4166 \pm 0.0056$ | $0.0000 \pm 0.0000$ | $0.0000 \pm 0.0000$ | $2.4166 \pm 0.0056$ | $0.0797 \pm 0.0002$ |
| TSA-PG(1.00) | Raspberry Pi | $3.9732 \pm 0.0146$ | $0.3257 \pm 0.0114$ | $0.0819 \pm 0.0026$ | $3.6485 \pm 0.0139$ | $0.9778 \pm 0.0014$ |
|  | MiniPC | $2.3817 \pm 0.0124$ | $0.0000 \pm 0.0000$ | $0.0000 \pm 0.0000$ | $2.3817 \pm 0.0124$ | $0.0785 \pm 0.0004$ |
| TSA-PG(1.25) | Raspberry Pi | $3.9866 \pm 0.0153$ | $0.2890 \pm 0.0118$ | $0.0727 \pm 0.0027$ | $3.6852 \pm 0.0141$ | $0.9770 \pm 0.0015$ |
|  | MiniPC | $2.3124 \pm 0.0149$ | $0.0000 \pm 0.0000$ | $0.0000 \pm 0.0000$ | $2.3124 \pm 0.0149$ | $0.0763 \pm 0.0005$ |
| TSA-PG(1.50) | Raspberry Pi | $3.9952 \pm 0.0184$ | $0.2719 \pm 0.0132$ | $0.0684 \pm 0.0031$ | $3.7022 \pm 0.0138$ | $0.9766 \pm 0.0016$ |
|  | MiniPC | $2.2800 \pm 0.0177$ | $0.0000 \pm 0.0000$ | $0.0000 \pm 0.0000$ | $2.2800 \pm 0.0177$ | $0.0752 \pm 0.0006$ |
| TSA-PG(1.75) | Raspberry Pi | $3.9825 \pm 0.0252$ | $0.2619 \pm 0.0143$ | $0.0659 \pm 0.0034$ | $3.7122 \pm 0.0147$ | $0.9765 \pm 0.0015$ |
|  | MiniPC | $2.2621 \pm 0.0223$ | $0.0000 \pm 0.0000$ | $0.0000 \pm 0.0000$ | $2.2621 \pm 0.0223$ | $0.0746 \pm 0.0007$ |
| TSA-PG(2.00) | Raspberry Pi | $3.9774 \pm 0.0163$ | $0.2797 \pm 0.0170$ | $0.0704 \pm 0.0041$ | $3.6944 \pm 0.0163$ | $0.9772 \pm 0.0016$ |
|  | MiniPC | $2.2981 \pm 0.0234$ | $0.0000 \pm 0.0000$ | $0.0000 \pm 0.0000$ | $2.2981 \pm 0.0234$ | $0.0758 \pm 0.0008$ |
| TSA-PG(2.50) | Raspberry Pi | $3.8085 \pm 0.0210$ | $0.3289 \pm 0.0202$ | $0.0807 \pm 0.0048$ | $3.7442 \pm 0.0234$ | $0.9818 \pm 0.0014$ |
|  | MiniPC | $2.2892 \pm 0.0369$ | $0.0000 \pm 0.0000$ | $0.0000 \pm 0.0000$ | $2.2892 \pm 0.0369$ | $0.0752 \pm 0.0013$ |

## E.3 HARD→EASY SCENARIO

Table 6: Per-machine metrics (hard→easy scenario).

| Method | Worker | Arrival rate | Loss rate | Loss ratio | Throughput | Utilization |
|---|---|---|---|---|---|---|
| UCB | Raspberry Pi | $3.9702 \pm 0.0225$ | $0.5074 \pm 0.0171$ | $0.1277 \pm 0.0037$ | $3.4646 \pm 0.0129$ | $0.9900 \pm 0.0012$ |
|  | MiniPC | $2.1280 \pm 0.0181$ | $0.0000 \pm 0.0000$ | $0.0000 \pm 0.0000$ | $2.1280 \pm 0.0181$ | $0.0761 \pm 0.0007$ |
| Baseline | Raspberry Pi | $3.9597 \pm 0.0207$ | $0.9045 \pm 0.0173$ | $0.2278 \pm 0.0031$ | $3.0652 \pm 0.0056$ | $0.9985 \pm 0.0006$ |
|  | MiniPC | $2.3609 \pm 0.0069$ | $0.0000 \pm 0.0000$ | $0.0000 \pm 0.0000$ | $2.3609 \pm 0.0069$ | $0.0901 \pm 0.0003$ |
| UCB-Mono | Raspberry Pi | $3.9702 \pm 0.0188$ | $0.4408 \pm 0.0153$ | $0.1109 \pm 0.0033$ | $3.5333 \pm 0.0083$ | $0.9740 \pm 0.0022$ |
|  | MiniPC | $2.5725 \pm 0.0117$ | $0.0000 \pm 0.0000$ | $0.0000 \pm 0.0000$ | $2.5725 \pm 0.0117$ | $0.0848 \pm 0.0004$ |
| TSA-PG(0.50) | Raspberry Pi | $3.9812 \pm 0.0169$ | $0.3503 \pm 0.0106$ | $0.0881 \pm 0.0023$ | $3.6238 \pm 0.0118$ | $0.9771 \pm 0.0018$ |
|  | MiniPC | $2.4160 \pm 0.0071$ | $0.0000 \pm 0.0000$ | $0.0000 \pm 0.0000$ | $2.4160 \pm 0.0071$ | $0.0797 \pm 0.0003$ |
| TSA-PG(0.75) | Raspberry Pi | $3.9689 \pm 0.0198$ | $0.3470 \pm 0.0112$ | $0.0873 \pm 0.0024$ | $3.6271 \pm 0.0117$ | $0.9769 \pm 0.0017$ |
|  | MiniPC | $2.4092 \pm 0.0073$ | $0.0000 \pm 0.0000$ | $0.0000 \pm 0.0000$ | $2.4092 \pm 0.0073$ | $0.0794 \pm 0.0003$ |
| TSA-PG(1.00) | Raspberry Pi | $3.9813 \pm 0.0159$ | $0.3223 \pm 0.0095$ | $0.0811 \pm 0.0021$ | $3.6518 \pm 0.0139$ | $0.9763 \pm 0.0017$ |
|  | MiniPC | $2.3625 \pm 0.0131$ | $0.0000 \pm 0.0000$ | $0.0000 \pm 0.0000$ | $2.3625 \pm 0.0131$ | $0.0779 \pm 0.0005$ |
| TSA-PG(1.25) | Raspberry Pi | $3.9763 \pm 0.0258$ | $0.2914 \pm 0.0126$ | $0.0733 \pm 0.0029$ | $3.6828 \pm 0.0145$ | $0.9748 \pm 0.0016$ |
|  | MiniPC | $2.2965 \pm 0.0162$ | $0.0000 \pm 0.0000$ | $0.0000 \pm 0.0000$ | $2.2965 \pm 0.0162$ | $0.0758 \pm 0.0006$ |
| TSA-PG(1.50) | Raspberry Pi | $3.9718 \pm 0.0175$ | $0.2780 \pm 0.0151$ | $0.0699 \pm 0.0035$ | $3.6961 \pm 0.0109$ | $0.9742 \pm 0.0017$ |
|  | MiniPC | $2.2701 \pm 0.0137$ | $0.0000 \pm 0.0000$ | $0.0000 \pm 0.0000$ | $2.2701 \pm 0.0137$ | $0.0748 \pm 0.0005$ |
| TSA-PG(1.75) | Raspberry Pi | $3.9752 \pm 0.0169$ | $0.2672 \pm 0.0164$ | $0.0672 \pm 0.0039$ | $3.7069 \pm 0.0136$ | $0.9747 \pm 0.0022$ |
|  | MiniPC | $2.2565 \pm 0.0260$ | $0.0000 \pm 0.0000$ | $0.0000 \pm 0.0000$ | $2.2565 \pm 0.0260$ | $0.0744 \pm 0.0009$ |
| TSA-PG(2.00) | Raspberry Pi | $3.9782 \pm 0.0185$ | $0.2820 \pm 0.0134$ | $0.0709 \pm 0.0031$ | $3.6921 \pm 0.0139$ | $0.9752 \pm 0.0017$ |
|  | MiniPC | $2.2846 \pm 0.0194$ | $0.0000 \pm 0.0000$ | $0.0000 \pm 0.0000$ | $2.2846 \pm 0.0194$ | $0.0754 \pm 0.0006$ |
| TSA-PG(2.50) | Raspberry Pi | $3.9731 \pm 0.0205$ | $0.3312 \pm 0.0196$ | $0.0813 \pm 0.0048$ | $3.7420 \pm 0.0269$ | $0.9802 \pm 0.0017$ |
|  | MiniPC | $2.2794 \pm 0.0382$ | $0.0000 \pm 0.0000$ | $0.0000 \pm 0.0000$ | $2.2794 \pm 0.0382$ | $0.0749 \pm 0.0013$ |

## F USE OF LLM

We used LLMs to polish writing. In addition, the tables produced by our experiments, in CSV, were exported to Latex using ChatGPT. The correctness of the result was manually verified.

## G    SOURCE CODE

An anonymized version of our source code repository is publicly available at: `https://anonymous.4open.science/r/Buffer_based_threshold-C531/README.md`

