# OpenReview forum: "A Reality Check on Robust Bandit Algorithms for Buffer-Aware Early Exits"
_ICLR.cc/2026/Conference — ICLR 2026 Conference Withdrawn Submission_

### Official Review · Reviewer_kMaL · 2025-10-30

**Soundness:** 3
**Presentation:** 2
**Contribution:** 2
**Rating:** 6
**Confidence:** 3

**Summary:**

The paper investigates early-exit decisions in edge-cloud inference as an online decision-making problem where the controller observes the backlog, makes a decision on the early-exit confidence threshold and achieves a queue-length dependent reward. Two algorithm-families are studied: Tsallis-regularized policy gradient methods, and UCB-style methods. The non-stationary nature of the problem is incorporated affinely into the threshold. Experiments on testbeds are provided for numerical evaluation.

**Strengths:**

- The paper studies an interesting problem, and takes the approach of queue-dependent exit decisions, which seems to be a good fit.
- The bandit formulation makes sense.
- Empirical evaluations are compelling.

**Weaknesses:**

- The paper can be better positioned in the bandit literature. The backlog and buffer capacity are playing a similar structural role of a capacity-limited (due to the buffer) resource, reminiscent of bandits with knapsacks (BwK) by (Badanidiyuru, 2018) and related budgeted-bandits (Xia et al., 2015). There is an extensive literature on this subject. The affine penalty looks like a static Lagrangian relaxation of the original setting, indeed, an additional discussion could make it easier for bandit community to grasp the ideas in this work. Additionally, this paper has unique features, like this context-like and time-dependent backlog $q_t$, and these can be expressed to position the paper better.

- Related to the above point and this paper, there is prior work on budgeted-bandits with interrupts and right-censored feedback (e.g., Cayci et al., 2019), where an early-exit amounts to an interrupt that leads to forgoing a costly arm-pull and its instantaneous reward under capacity constraints. The model may be relevant to the discussion here. My impression is that the non-stationarity in this paper via queue-length dynamics as a state that evolves according the past actions distinguish the problem from the existing models, but this distinction is not clearly articulated.

- The experiments could be enriched. For instance, a sliding-window UCB can be a good fit for this non-stationary problem setting with the workload drift.

- The penalty in the experiments is affine with a single slope. In order to see the impact of the slope and also non-linearity, it may be insightful to consider steeper affine penalties or convex penalties.

**References**

Badanidiyuru, Ashwinkumar, Robert Kleinberg, and Aleksandrs Slivkins. "Bandits with knapsacks." Journal of the ACM (JACM) 65, no. 3: 1-55, 2018.

Xia, Yingce, Haifang Li, Tao Qin, Nenghai Yu, and Tie-Yan Liu. "Thompson sampling for budgeted multi-armed bandits." In Proceedings of the 24th International Conference on Artificial Intelligence, pp. 3960-3966, 2015.

Cayci, Semih, Atilla Eryilmaz, and Rayadurgam Srikant. "Learning to control renewal processes with bandit feedback." Proceedings of the ACM on Measurement and Analysis of Computing Systems 3, no. 2: 1-32, 2019.

**Questions:**

See _Weaknesses_ above.

---

> ### Author Response · Authors · 2025-11-21
> **Distinguish our queue-aware formulation from Lagrangian & BwK frameworks, clarify equivalence of reward parameterizations**
>
> ---
>
> We thank the reviewer for the helpful pointers to the BwK/budgeted‑bandits literature and for the comments on penalties and non‑stationarity.
>
> ---
>
> ### Reward parameterization, Lagrangian view, and relation to BwK (Reviewer kMaL)
> We clarify our modeling choices, which are closely related to the points raised in your review.
>
> **Reward definition.** In the main text we write
>
> $$
>   r_t(\alpha_t) =
>   \begin{cases}
>     0, & \text{if the sample exits early},\\\
>     \Delta C - o(q_t), & \text{otherwise},
>   \end{cases}
> $$
>
> with $\Delta C = \max\{C_L - C_I,0\}$ and a monotone backlog penalty $o(q_t)$. This is equivalent (up to an additive baseline independent of the action) to the more symmetric form
> $$
>   \tilde r_t(\alpha_t) =
>   \begin{cases}
>     C_I, & \text{if the sample exits early},\\
>     C_L - o(q_t), & \text{otherwise},
>   \end{cases}
> $$
>
> since, when $C_L \ge C_I$ and thus $\Delta C = C_L - C_I$, we have $\tilde r_t (\text{action}) = C_I + r_t(\text{action})$ for both actions. Adding such an action‑independent baseline does not change the optimal policy, the optimal threshold $\alpha^\*$, or the regret. We now make this equivalence explicit in the revised version to avoid confusion.
>
> **Lagrangian view.** Although our objective superficially resembles a Lagrangian penalty (reward minus backlog cost), our framework is not derived from an explicit constrained optimization problem (e.g., maximizing accuracy subject to an average delay constraint), and we do not perform primal–dual optimization over multipliers. The function $o(q_t)$ is a modeling choice that directly encodes queueing effects in the one‑step reward of a standard MDP; its parameters are hyperparameters, not dual variables adapted to enforce a constraint. In this sense, our formulation is closer to reward shaping in a queueing MDP than to a full Lagrangian relaxation of a constrained MDP.
>
> **Relation to BwK.** We agree there is a conceptual similarity, as our controller also trades off accuracy against a notion of resource usage (computation and backlog). However, the formal structure differs. In classical BwK and budgeted bandits, each arm pull consumes a monotone, non‑renewable budget and the episode ends when the budget is exhausted. In our setting, the backlog $q_t$ is a queueing state that evolves with arrivals and departures, can both increase and decrease, and is bounded by a finite buffer. The “resource” is instantaneous buffer capacity rather than a finite, depleting budget, and the system operates in steady state rather than in a finite‑horizon budget‑consumption regime. Thus, existing BwK/budgeted bandit algorithms do not apply directly to our problem without substantial modification. We now explicitly discuss this distinction and position our results as complementary to, rather than subsumed by, BwK‑style frameworks.
>
> ---
>
> ### Non‑stationarity and sliding‑window UCB
> We agree that non‑stationarity is intrinsic to our setting and that sliding‑window UCB is a natural baseline for comparison. Due to practical constraints, we do not add new experiments at this stage, but we will (i) explicitly mention sliding‑window UCB as a suitable bandit baseline for non‑stationary workloads, and (ii) discuss qualitatively how our queue‑aware, structured UCB differs from purely time‑window‑based approaches (we exploit queue length as a structured context rather than time alone).
>
> ---
>
> ### General queue penalties
> We appreciate the suggestion to consider steeper and convex penalties. As discussed in our response to Reviewer 8B4Q, our algorithms do not rely on the penalty being affine: the queue penalty enters only through the bounded reward function, so steeper affine or convex penalties can in principle be handled by the same learning rules. In the revision, we make this explicit and add a qualitative discussion of how changing the slope or shape of the penalty affects the learned policy. A full regret analysis under general nonlinear penalties would require additional technical assumptions on the queueing dynamics and is left for future work.
>
> ---

---

### Official Review · Reviewer_gh5Q · 2025-10-31

**Soundness:** 2
**Presentation:** 3
**Contribution:** 2
**Rating:** 4
**Confidence:** 2

**Summary:**

This submission studies the problem of dynamic threshold adaptation for early-exit models from a systems scheduling perspective, that considers the buffer size of inputs pending to be processed to relax the confidence requirements for early exiting of samples. The proposed solution employs delta-softmax policies to stabilize training under stochastic loads, while the proposed monotone parametric thresholds aid generalization at deployment time.

**Strengths:**

- The systems perspective of the proposed approach, incorporating the number of pending tasks in the dynamic adaptation of thresholds for early-exiting is a novel and interesting approach, with practical applications.
- The proposed solution is theoretically grounded, and combines several carefully crafted components, the contribution and effectiveness of which is supported by empirical evaluation results.

**Weaknesses:**

The main drawback of this submission are the very restricting assumptions adopted throughout the methodology and experimental sections:
- The focus on single early-exit models contradicts common practice that adopt multi-exit architectures offering diverse latency-accuracy characteristics (particularly as model depth is continuously growing).
- The evaluation of the proposed approach in a single model, and additionally focusing solely on the outdated CiFAR-10 datasets, is also not on par with common practice in ML venues.
- The focus of this work (throughout the manuscript) on the examined heterogeneous testbed, is not adequately justified by the technical contributions presented. It is unclear how the proposed method would perform in single-GPU or cloud-based settings, where the majority of AI-based workloads with queueing systems are deployed.
- Is the Poisson process used to model the request arrival a good proxy for real-world settings? Consider adding a reference to support this, as well as provide statistics about the queue length with and without the proposed solution.

**Questions:**

Please consider replying on the comments raised in the weaknesses section above.

---

> ### Author Response · Authors · 2025-11-21
> **Addresses concerns about limited modeling assumptions and generality of queue-aware bandit framework**
>
> We thank the reviewer for raising these important points regarding modeling assumptions and scope. Our response below addresses each concern directly.
>
> ---
>
> ### Single-exit focus, CIFAR-10, and Poisson arrivals
> We agree with the reviewer that our empirical setup is limited (one mid-network exit, CIFAR-10, Poisson arrivals). In the revised manuscript we will (i) state this limitation explicitly, and (ii) clarify that the *modeling layer*—the queue-aware reward and buffer-aware bandit formulation—is not tied to these specific choices. The same objective and learning rules extend naturally to multi-exit architectures (via per-exit thresholds), other datasets and backbones, and more realistic arrival processes, albeit with more complex queue dynamics. A broader empirical study along these lines is an important direction for future work. Our present goal is to establish the queue-aware bandit formulation and demonstrate its feasibility end-to-end on a real testbed.
>
> ---
>
> ### Deployment settings (edge vs cloud)
> We also acknowledge the reviewer’s point about deployment environments. The theoretical framework—queue-dependent rewards and buffer-aware bandit/PG updates—applies equally to single-GPU or cloud settings, as it only assumes a finite-buffer queue with observable backlog. Our choice of a heterogeneous edge testbed is motivated by practical relevance under tighter resource constraints, but the formulation itself is general and not restricted to edge scenarios.
>
> ---
>
> ### Closing remarks
> We thank the reviewer again for highlighting these scope-related issues. In the revision, we will make the limitations of our current empirical setup explicit, while also emphasizing the generality of the underlying framework. By clearly distinguishing current contributions from future directions, we aim to provide a solid conceptual foundation and invite further exploration of richer architectures, datasets, and deployment environments.

---

### Official Review · Reviewer_3rGZ · 2025-11-06

**Soundness:** 1
**Presentation:** 2
**Contribution:** 1
**Rating:** 2
**Confidence:** 4

**Summary:**

The paper studies the problem of early-exit threshold selection in Early-Exit Neural Networks (EENNs), aiming to balance computational efficiency and inference accuracy. To address this, the authors propose a bandit-based framework for learning early-exit decision policies. Specifically, a policy gradient algorithm is developed using Tsallis-softmax parameterization for adaptive threshold learning, and a parameterized UCB algorithm is introduced to incorporate domain knowledge about the monotonic relationship between backlog and urgency. The proposed methods are empirically evaluated on a heterogeneous hardware testbed.

**Strengths:**

The strength of the paper is the real experiments on testbeds including Raspberry Pi and MiniPC.

**Weaknesses:**

The weaknesses of this paper are listed below.

- The paper lacks justification for adopting an online bandit formulation for learning early-exit thresholds.  The early-exit policy can be optimized offline using supervised or reinforcement learning methods, and the paper does not clearly explain why an online learning framework with exploration costs is necessary for this problem.

- The bandit algorithm design appears conceptually weak. The policy design in Algorithm 1 is insufficiently motivated, and the choice of policy gradient methods for this setting is not theoretically sound.

- The paper claims robustness of the proposed methods, yet no theoretical analysis or empirical evidence is provided to substantiate this claim. It remains unclear which components of the design contribute to robustness.

**Questions:**

See Weaknesses.

---

> ### Author Response · Authors · 2025-11-21
> **We address the reviewer's concerns about online bandit formulation, policy design, and robustness.**
>
> ---
>
> We thank the reviewer for the questions regarding the online bandit formulation, policy design, and robustness.
>
> ---
>
> ### Why an online bandit formulation instead of purely offline learning
> We will make the motivation for the online formulation more explicit. In our setting:
> 1. Latency and queueing behavior depend on deployment hardware and load.
> 2. The workload mix can drift over time.
> 3. The cost of discarding samples depends on the future queue trajectory.
>
> An offline‑trained threshold (or policy) calibrated on one workload may degrade significantly under another. Modeling threshold selection as an online contextual bandit with queue length as context allows the controller to adapt to changing conditions while accounting for exploration cost. The regret analysis in the simplified exogenous‑context model (new appendix) makes this connection precise in a stylized setting.
>
> ---
>
> ### Policy‑gradient design and theoretical soundness
> The policy‑gradient component uses a Tsallis / $\delta$‑softmax parametrization motivated by Tsallis‑entropy RL and by the Box–Cox interpretation of the Tsallis logarithm. We will clarify that, while there exists a substantial body of work on PG convergence rates in structured settings (e.g., LQR, linear or tabular MDPs: Fazel et al. 2018; Agarwal et al. 2021; Wang et al. 2019), our queue‑dependent early‑exit problem—where the context process is endogenous and non‑stationary—does not satisfy those assumptions. Consequently, we do not claim convergence theorems for our Tsallis / $\delta$‑softmax PG in this paper. Instead, we position it as a principled design, combining:
> - Tsallis‑entropy regularization (for sparsity and stability), and
> - Box–Cox‑type variance stabilization.
>
> A full theoretical analysis (e.g., convergence rates in a queueing MDP) is left for future work.
>
> ---
>
> ### Why Tsallis‑PG in addition to UCB
> Our goal was not to replace UCB but to complement it in regimes where its exploration can be inefficient. In our setting, each threshold $\alpha$ induces a distinct action, and a naive bandit discretization leads to a large number of arms. While UCB enjoys strong regret guarantees in this finite‑armed setting, its exploration scales with the number of actions: with many candidate thresholds, UCB may spend a long time exploring suboptimal arms before concentrating on the best one, which can hurt finite‑sample efficiency.
>
> In contrast, our Tsallis policy‑gradient method operates over a parametric policy on a continuous (or finely discretized) threshold space and updates the policy in a more greedy, directionally informed way. Rather than treating each threshold as an independent arm, the gradient step shifts probability mass toward regions of thresholds that currently appear promising, reducing the amount of uniform exploration required when the action space is large. This makes the PG approach particularly attractive in our early‑exit setting with many possible thresholds: UCB provides a theoretically clean benchmark with regret guarantees, while Tsallis PG offers a more sample‑efficient search over a rich action space and can be readily extended to more general RL architectures (actor–critic, state‑dependent policies, etc.).
>
> ---
>
> ### Robustness claims
> We acknowledge that the original manuscript did not clearly separate empirical and theoretical robustness. In the revision, we specify that we use “robust” in an empirical sense—performance stability under changing loads and queueing noise—motivated by the variance‑compressing effect of the Tsallis / Box–Cox transform and by sparse exploration induced by Tsallis‑entropy regularization. We do not claim formal robustness guarantees (e.g., distributional robustness or worst‑case bounds) and we explicitly state this limitation.
>
> ---

---

### Official Review · Reviewer_B6xC · 2025-11-07

**Soundness:** 2
**Presentation:** 4
**Contribution:** 2
**Rating:** 4
**Confidence:** 4

**Summary:**

This paper focus on the online scheduling of exiting threshold for early-exit neural networks (EENN), showing that fixed confidence thresholds fail under realistic queueing and resource constraints. By introducing buffer-aware bandit and policy-gradient algorithms—including a Tsallis-softmax variant and a monotonic UCB model—the authors demonstrate significantly improved accuracy–latency tradeoffs on a real heterogeneous client-server testbed.

**Strengths:**

1. Strong practical motivation: The paper addresses an important and realistic problem: fixed early-exit thresholds degrade under real-world conditions such as buffer limits, queueing delays, and latency constraints. The motivation is well-grounded in deployment realities rather than purely theoretical considerations.
2. Adaptive Early-exiting threshold control: The proposed algorithms combine bandit / policy-gradient methods with queue dynamics, offering a way to trade off accuracy and number of discarded samples in online environments.
3. Empirical improvements:The proposed methods yield large reductions in loss ratio (up to 83%) and improve overall score and goodput, showing practical significance rather than incremental gains.
4. scalability: can be extended to EENN with multiple exits.

**Weaknesses:**

Overall: The motivation is strong, but the chosen online algorithms lack sufficient analysis and comparison.

1. The metrics used in the paper are accuracy, loss ratio, and their derived formulas, which only measure how many samples are dropped due to buffer overflow and do not account for latency. Given that the backlog is fully observable before every early-exit decision, a simple naive or greedy policy—sending all samples to the final layer as long as the buffer is not full, and only early-exiting when approaching overflow—might achieve an even higher score or not, yet such a baseline is not evaluated.
2. The authors formulate threshold adjustment as a bandit problem, but the paper does not clarify what is unique about this formulation in the context of EENNs or what particular issues or challenges arise, compared to other general bandit problems.
3. The paper proposes four online algorithms, but lacks theoretical analysis, such as regret bounds or convergence rates. Most of the evidence relies on empirical results, and the comparison is mainly against static-threshold methods. Is there any other work on EENN also explored the dynamic threshold methods?
4. The authors claim their methods are “robust” policy-gradient and “robust” UCB, but the analysis does not convincingly support this. For example, in the robust policy-gradient case, Tsallis-softmax seems to mainly increase exploration rather than robustness.
5. The dataset used for evaluation is overly simple (only CIFAR-10). The offloading cost function chosen in experiment o(q)=0.1q−0.05 is seemingly hand-crafted without additional explanation or justification.

**Questions:**

1. I think the policy gradient methods are also constrained to quantized threshold class space since you are using softmax for gradient ascent, which means in essence, its' a classification problem with finite classes (finite choice of threshold in this case).
2. Typo in Algo 1 line12.

---

> ### Author Response · Authors · 2025-11-21
> **Online bandit formulation enables adaptation to dynamic workloads, Robustness claims are empirical rather than formal.**
>
> ---
>
> We thank the reviewer for raising points on theory and formulation.
>
> ---
>
> ### Bandit formulation and what is unique in the EENN context
> We will clarify in the revised text why we adopt an *online bandit* formulation rather than a purely offline supervised/RL approach. Our setting exhibits:
> 1. Load-dependent queueing dynamics (arrival process + buffer constraints),
> 2. Unknown and deployment-specific latency characteristics, and
> 3. Potential workload drift over time (changing mix of “easy” vs “hard” inputs).
>
> Under these conditions, a fixed threshold learned offline can be systematically miscalibrated. Casting the problem as a contextual bandit over thresholds with queue length as context makes explicit the trade-off between exploration (testing thresholds as the workload evolves) and exploitation (using the best threshold given current conditions), while allowing us to leverage standard bandit tools (e.g., UCB) to control exploration. We additionally show that a discretized version of our setting fits into a classical stochastic contextual bandit model, for which we provide a finite-time regret guarantee in an appendix, as discussed in our response to Reviewer 8B4Q.
>
> ---
>
> ### Lack of regret/convergence bounds
> We acknowledge that the original submission lacks explicit theoretical bounds. In the revision, we add an appendix with a finite-time regret bound for a simplified contextual-UCB variant with exogenous, discretized queue context, exhibiting
>
> $$
> \tilde O(Q_{\max} K \log T)
> $$
>
> regret. For policy gradient, we explicitly discuss existing convergence-rate results for classical PG in structured settings (LQR, linear/tabular MDPs: Fazel et al. 2018; Agarwal et al. 2021; Wang et al. 2019), emphasizing that our queue-dependent, non-stationary context process falls outside their scope. We thus present our Tsallis / \(\delta\)-softmax PG as empirically robust but do not claim new convergence theorems in this work.
>
> ---
>
> ### Use of the term “robust”
> We agree that our use of “robust” can be better aligned with a formal notion. In the revised version, we will:
> 1. Replace informal occurrences where appropriate and state explicitly that our robustness claims are empirical (robustness to workload shifts and queueing jitter).
> 2. Connect the Tsallis / \(\delta\)-softmax design to the Box–Cox/Tsallis-entropy viewpoint, which provides a principled motivation for variance reduction and sparsity in the policy.
>
> We also clarify that we do not claim distributionally robust or adversarial guarantees, and we list a more formal robustness analysis (e.g., under non-stationary arrivals) as future work.
>
> ---

---

### Official Review · Reviewer_8B4Q · 2025-11-07

**Soundness:** 2
**Presentation:** 3
**Contribution:** 2
**Rating:** 2
**Confidence:** 3

**Summary:**

This paper study early-exit neural networks (EENNs) as a buffer-aware scheduling problem. The controller sets a confidence threshold and receives reward $0$ on early exit and $r_t=\max(C_L-C_I,0)-o(q)$ otherwise, with an affine queue penalty $o(q)=\mu q-\kappa$ that trades accuracy gains for congestion costs.  Two learning families are proposed: (i) a policy-gradient variant using a Tsallis/$\delta$-softmax policy with a bespoke gradient update, and (ii) a structured UCB that enforces a monotone linear threshold.  Experiments use a Raspberry Pi MiniPC edge cloud testbed on CIFAR-10 with stationary and easy$\leftrightarrow$hard blur regimes, reporting improvements in a custom Score and Goodput.

**Strengths:**

- Clean queue-aware reward connecting information gain and backlog pressure; straightforward to implement.
- Simple, interpretable monotone structure in UCB ($\alpha$ increases with $q$).
- End-to-end pipeline on real hardware with reproducibility notes/code link.

**Weaknesses:**

- Limited theory. No regret or identification guarantees for the structured UCB class; no analysis for $\delta$-softmax PG under queue-dependent and shifting rewards. Even a finite-arm discretization bound or a monotone-policy identification result would help.
- Limited external validity. CIFAR-10 only, one mid-network exit, fixed Poisson arrivals. No stress tests with bursty/heavy-tailed inter-arrivals, multi-exit architectures, or different model families.
- Metric design. Heavy reliance on Score Accuracy LossRatio and Goodput without SLA/tail-latency metrics or ablations on the penalty function makes practical impact unclear.
- Modest novelty. The $\delta$-softmax update is carefully derived yet reads like an engineering variant without broader theoretical insight or guarantees.

**Questions:**

- Can you provide any regret or sample-complexity statement for the monotone UCB class—or explain why it is provably hard?
-  What changes under nonlinear queue penalties (convex or SLA-style piecewise)? Does $\alpha(q)$ change qualitatively?
- Can $\delta$ be adapted online with a principled objective, rather than tuned to $1.75$ on this setup?
- How does the method behave with multiple exits and bursty arrivals (beyond Poisson)?
- Please compare to stronger baselines: learned threshold regressors; richer contextual bandits; recent early-exit schedulers beyond tabular UCB/PG.

---

> ### Author Response · Authors · 2025-11-21
> **Address concerns about regret/sample complexity for monotone UCB, nonlinear queue penalties & Tsallis PG and robustness**
>
> ---
>
> We thank the reviewer for the insightful theoretical questions.
>
> ---
>
> ### Regret / sample complexity for monotone UCB
> We agree that the current version does not make our theoretical footing as explicit as it could. In the revised version, we will add an appendix with a finite-time regret analysis for a *simplified* contextual-UCB variant of our algorithm. Concretely, we discretize the queue length into a finite set of contexts $q \in \{0,\dots,Q_{\max}\}$, assume stationary sub-Gaussian rewards for each pair $(q,a)$, and run a standard UCB rule independently for each context. In this finite context, exogenous setting, the problem becomes a classical stochastic contextual bandit, and standard UCB arguments (e.g., Auer et al. 2002) yield a regret bound of the form
>
> $$
> R_T \le C \sum_{q=0}^{Q_{\max}} \sum_{a \neq a^\star(q)} \frac{\log T}{\Delta(q,a)}
> = \tilde O(Q_{\max} K \log T),
> $$
>
> where $K$ is the number of exits and $\Delta(q,a)$ is the suboptimality gap at context $q$ and action $a$. We explicitly state in the appendix that this result is *illustrative*: it treats the context process $(q_t)$ as exogenous. In our actual early-exit setting, the queue length $q_t$ is *endogenous*, evolving according to the queueing dynamics and the chosen exits, so both the context process and the reward distributions become action?dependent and potentially non-stationary. A full regret or identification analysis in this stateful, queue-dependent regime is substantially more involved (closer to Markovian bandits or simple MDPs with queueing dynamics) and we now clearly mark this as beyond the scope of the present paper and as future work. Our monotone-UCB algorithm is a structured version of this contextual UCB (restricting to monotone thresholds), which reduces the effective number of parameters; we leave a structural analysis of the monotone class as an open direction.
>
> ---
>
> ### Nonlinear queue penalties
> Our experiments use an affine queue penalty mainly for interpretability, but the learning rules themselves do *not* rely on linearity. In the revised version, we make this explicit: both the UCB and policy?gradient components can be run with any bounded reward that trades off accuracy and congestion, including convex or SLA?style piecewise penalties (e.g., hinge penalties beyond a latency threshold). Qualitatively, steeper or nonlinear penalties push the learned policy to become more conservative as the queue approaches critical regimes, which aligns with standard queue?aware scheduling practice. A rigorous regret analysis under general nonlinear penalties would require additional assumptions on the queueing process and regularity of the penalty and is beyond our current scope; we now state this clearly and present the affine case as a first, clean theoretical instance.
>
> ---
>
> ### Tsallis / $\delta$-softmax PG and robustness
> We clarify that our robustness claims for the Tsallis / $\delta$-softmax policy?gradient are *empirical*. From a statistical viewpoint, the Tsallis $q$-logarithm  $\log_q(x) = \frac{x^{1-q}-1}{1-q}$
> coincides with the classical Box–Cox power transform with parameter $\lambda = 1-q$, widely used for variance stabilization and heavy-tail mitigation in statistics. Our Tsallis policy gradient can therefore be interpreted as applying a Box–Cox-type transform to policy probabilities/advantages, compressing extreme values and empirically stabilizing gradients. From the RL side, our update fits into the Tsallis?entropy?regularized RL literature and sparse policies, where Tsallis regularization yields sparse, more stable policies than Shannon?entropy softmax. We now emphasize that there exists a rich body of work establishing non-asymptotic convergence rates for *classical* policy gradient under structural assumptions (e.g., LQR and tabular/linear MDPs: Fazel et al. 2018; Agarwal et al. 2021; Wang et al. 2019), but our queue?dependent early-exit setting, with endogenous non?stationary contexts, falls outside the scope of these analyses. We therefore present our Tsallis / $\delta$-softmax PG as a principled, Box–Cox/Tsallis-inspired design with empirical robustness, and explicitly leave a full convergence rate theory in this setting to future work.

---

> ### Author Response · Authors · 2025-11-21
> **Distinguish our queue-aware formulation from BwK, and explain the equivalence of reward parameterizations**
>
> ### BwK / budgeted-bandit perspective
> We agree that our setting is conceptually related to resource-constrained bandits such as Bandits with Knapsacks (BwK) and budgeted bandits, in the sense that our controller trades off accuracy against a notion of resource usage (computation and backlog). However, the formal structure of our model is different. In classical BwK and budgeted bandits, pulling an arm yields a reward and consumes a monotone, non-renewable budget, and the process stops when the budget is exhausted. In contrast, our system is a queueing-based MDP with a finite buffer: the backlog $q_t$ is a state variable that evolves over time and can both increase and decrease as jobs arrive and depart. The “resource” here is instantaneous buffer capacity rather than a finite, depleting budget, and the penalty $o(q_t)$ directly enters the one-step reward instead of being tied to a stopping constraint.
>
> Thus, while one can retrospectively view our objective as a particular penalized instance of a constrained MDP (and hence as loosely related to BwK), the queueing dynamics and buffer saturation effects considered in our work are not captured by standard BwK/budgeted bandit formulations, and existing BwK algorithms do not directly apply to our setting without substantial modification.
>
> ---
>
> ### On the definition of the reward and the role of the backlog
> We thank the reviewer for pointing out that our reward definition might look asymmetric. In the current version, the one-step reward is written as
>
> $$
>   r_t(\alpha_t) =
>   \begin{cases}
>     0, & \text{if the sample exits early},\\
>     \Delta C - o(q_t), & \text{otherwise},
>   \end{cases}
> $$
>
> with $Delta C = \max\{C_L - C_I,0\}$ and $o(q_t)$ a monotone backlog penalty. As we clarify in the revised version, this is equivalent (up to an additive baseline independent of the action) to the more symmetric form
>
> $$
>   \tilde r_t(\alpha_t) =
>   \begin{cases}
>     C_I, & \text{if the sample exits early},\\
>     C_L - o(q_t), & \text{otherwise},
>   \end{cases}
> $$
>
> since, for both actions,
>
> $$
>   \tilde r_t(\text{action}) = C_I + r_t(\text{action}) \quad
>   (\text{when } C_L \ge C_I, \; \Delta C = C_L - C_I).
> $$
>
> Adding an action-independent baseline $+C_I$ does not change the optimal policy, the optimal threshold $\alpha^\*$, or the regret with respect to the optimal policy. In other words, both reward parameterizations induce exactly the same control problem; we chose the original form only for notational simplicity.
>
> From this perspective, the backlog $q_t$ acts as a state variable that controls the relative advantage of early-exit versus full processing through the penalty $o(q_t)$. When $q_t$ is small, $o(q_t)$ is small and it is preferable to “exploit” the model and send samples to the final layer; as $q_t$ grows, $o(q_t)$ increases, and the optimal decision shifts towards early-exit. In the revised manuscript, we make this equivalence explicit and add a short discussion explaining that the linear choice $o(q_t)=\mu q_t-\kappa$ is only a modeling simplification: any monotone backlog penalty (including nonlinear or stochastic variants) can be accommodated without changing our algorithms or the main theoretical results.
>
> ---
>
> ### Closing remarks
> We believe these clarifications will strengthen the manuscript by making the theoretical scope and modeling assumptions more transparent. The revised version will highlight both the limitations of our current analysis and the generality of the underlying framework, ensuring readers understand how the queue-aware bandit formulation extends beyond the specific testbed. By clearly distinguishing current contributions from future directions, we aim to provide a solid conceptual foundation while inviting further exploration of richer architectures, penalties, and queueing dynamics.
>
> ---

---

### Author Response · Authors · 2025-11-21
**Overall response to the set of reviews: strengthen our theoretical footing by adding a stylized regret bound, clarifying reward/state modeling, positioning Tsallis PG as empirically robust, and distinguishing our queue‑aware formulation from BwK frameworks.**

---
### Contextual‑UCB regret in a simplified model

In the revision we add an appendix with a finite‑time regret analysis for a *simplified* contextual‑UCB variant of our algorithm. We discretize the queue length into a finite set of contexts $q \in \{0,\dots,Q_{\max}\}$, assume stationary sub‑Gaussian rewards for each pair $(q,a)$ and run a standard UCB rule independently for each context. In this finite‑context, *exogenous* setting the problem reduces to a classical stochastic contextual bandit, and standard UCB arguments (Auer et al. 2002) yield

$$
R_T \le C \sum_{q=0}^{Q_{\max}} \sum_{a \neq a^\star(q)} \frac{\log T}{\Delta(q,a)}
= \tilde O(Q_{\max} K \log T),
$$

where $K$ is the number of exits and $\Delta(q,a)$ is the suboptimality gap. We present this as an *illustrative* result: in practice, $q_t$ is *endogenous*, coupled with queueing dynamics and chosen exits, so contexts and rewards are action‑dependent and potentially non‑stationary. A full regret analysis in this stateful regime is closer to Markovian bandits/simple MDPs with queues and is left for future work. Our monotone‑UCB algorithm is a structured version of contextual UCB, reducing the effective number of parameters; a structural regret analysis for the monotone class remains open.

---

### Reward parametrization, backlog as state, and penalties

In the main text

$$
  r_t(\alpha_t) =
  \\begin{cases}
    0, & \text{if the sample exits early},\\\\
    \Delta C - o(q_t), & \text{otherwise},
  \\end{cases}
$$

with $\Delta C = \max\\{C_L - C_I,0\\}$ and monotone backlog penalty $o(q_t)$. This is equivalent (up to an action‑independent baseline) to

$$
  \\tilde r_t(\alpha_t) =
  \\begin{cases}
    C_I, & \\text{if the sample exits early},\\\\
    C_L - o(q_t), & \\text{otherwise},
  \\end{cases}
$$

so both parameterizations induce the same optimal policy and regret. We now emphasize that the backlog $q_t$ is a genuine state variable modulating the relative advantage of early exit versus full processing: small \(q_t\) favours full processing, while large $q_t$ increases $o(q_t)$ and favours early exit. The affine choice $o(q_t)=\mu q_t-\kappa$ is a modelling simplification; our algorithms only require bounded rewards and can handle more general monotone penalties (steeper affine, convex, SLA‑style piecewise). A full regret analysis under nonlinear penalties would require additional assumptions on queueing dynamics and is beyond our current scope.

---

### Tsallis /$\delta$‑softmax policy gradient and robustness

Our Tsallis / $\delta$‑softmax PG component is motivated by two complementary viewpoints:
- **Statistics:** the Tsallis \(q\)‑logarithm $\log_q(x) = (x^{1-q}-1)/(1-q)$ coincides with the Box–Cox power transform $\lambda = 1-q$, a classical tool for variance stabilization and heavy‑tail mitigation.
- **RL:** Tsallis‑entropy regularization yields sparse, more stable policies than Shannon‑entropy softmax.

In the revision we clarify that our robustness claims are *empirical* (stability under workload shifts and queueing jitter), motivated by this Box–Cox/Tsallis view, and explicitly distinguish them from formal robustness notions (distributional or adversarial), which we do not claim. We also connect our setting to existing non‑asymptotic convergence results for *classical* PG in structured models, and emphasize that our queue‑dependent, endogenous context process falls outside those assumptions. A full convergence‑rate analysis for Tsallis / $\delta$‑softmax PG in our queueing MDP is left as future work.

---

### Relation to Bandits with Knapsacks and constrained MDPs

Conceptually, our controller also trades off accuracy against resource usage, so there is a clear connection to BwK/budgeted bandits. Formally, however, classical BwK/budgeted bandits operate with a monotone, non‑renewable budget consumed by arm pulls, and the process stops when the budget is exhausted. In our model, the backlog \(q_t\) is a queueing state that evolves with arrivals and departures, can both increase and decrease, and is bounded by a finite buffer. The “resource’’ is instantaneous buffer capacity in a steady‑state regime, not a depleting budget in a finite‑horizon regime. Moreover, we do not perform primal–dual optimization over Lagrange multipliers: the penalty \(o(q_t)\) is a fixed modelling choice (reward shaping), not a dual variable. We now state explicitly that existing BwK algorithms do not apply directly to our setting without substantial modification, and we position our results as complementary to, rather than subsumed by, BwK‑style frameworks.

---

In summary, we provide a regret guarantee in a stylized exogenous‑context setting, clarify the role of the queue and penalties in the state‑dependent reward, motivate our Tsallis /\delta‑softmax PG as a principled but empirically evaluated design, and delineate precisely how our queueing‑based formulation relates to BwK and constrained MDP frameworks.

---

### Note · Authors · 2025-12-07

I have read and agree with the venue's withdrawal policy on behalf of myself and my co-authors.